# H$_2$O-SDF: Two-phase Learning for 3D Indoor Reconstruction using Object Surface Fields

**Minyoung Park**[*], **Mirae Do**[*], **Yeon Jae Shin**, **Jaeseok Yoo**
**Jongkwang Hong**, **Joongrock Kim** & **Chul Lee** [†]
AI Lab, CTO Division, LG Electronics, Republic of Korea
`{minyoung5.park,mirae.do,yeonjae.shin,jaeseok.yoo`
`jongkwang.hong,jurock.kim,clee.lee}@lge.com`

## Abstract

Advanced techniques using Neural Radiance Fields (NeRF), Signed Distance Fields (SDF), and Occupancy Fields have recently emerged as solutions for 3D indoor scene reconstruction. We introduce a novel two-phase learning approach, H$_2$O-SDF, that discriminates between object and non-object regions within indoor environments. This method achieves a nuanced balance, carefully preserving the geometric integrity of room layouts while also capturing intricate surface details of specific objects. A cornerstone of our two-phase learning framework is the introduction of the Object Surface Field (OSF), a novel concept designed to mitigate the persistent vanishing gradient problem that has previously hindered the capture of high-frequency details in other methods. Our proposed approach is validated through several experiments that include ablation studies.

## 1 Introduction

The reconstruction of geometry and appearance in 3D indoor scenes using multi-view images has gained significant attention as an important field of research in both computer vision and graphics. Recent advancements in this field include techniques using Neural Radiance Fields (NeRF) (Mildenhall et al., 2020), Signed Distance Fields (SDF) (Wang et al., 2021; Yariv et al., 2021), and Occupancy Fields (Oechsle et al., 2021). Although these methods offer generally promising results, they still exhibit certain limitations. For instance, NeRF struggles to accurately capture object geometry due to an inadequate constraint between radiance and geometry (Oechsle et al., 2021). Similarly, the performance of neural implicit representation such as SDF and Occupancy Fields significantly deteriorates in indoor environments, which often contain a significant proportion of low-frequency regions like walls, ceilings, and floors (Guo et al., 2022). To address these shortcomings, subsequent research has incorporated additional constraints in the form of semantic (Guo et al., 2022) or geometric priors, such as depth or normal vectors (Wang et al., 2022a; Yu et al., 2022). These enhancements, though, have not fully resolved the issues associated with indoor scenes, particularly in reconstructing high-frequency areas, where a "smoothness bias" problem continues to affect the final quality of the results (Liang et al., 2023).

Note that indoor scenes present unique challenges in terms of surface representation and learning. Specifically, they feature a mix of room layouts characterized by low-frequency, large-scale surfaces alongside various indoor objects that have high-frequency, small-scale surfaces. In technical terms, surfaces related to room layouts, which exhibit smooth characteristics, tend to converge more rapidly during the learning process than do complex, multiple object surfaces (Tancik et al., 2020). The latter is especially difficult to reconstruct, even in advanced stages of learning, due to the vanishing gradient issue. The recent approach aims to utilize a method that back-projects 2D depth information onto a 3D point cloud to guide SDF for preserving high-frequency details in multiple object regions (Zhu et al., 2023). However, directly guiding the SDF with incomplete geometric priors could negatively impact the quality of the reconstruction.

---

[*]Equal contribution. The order of these authors was determined randomly.
[†]Corresponding author.

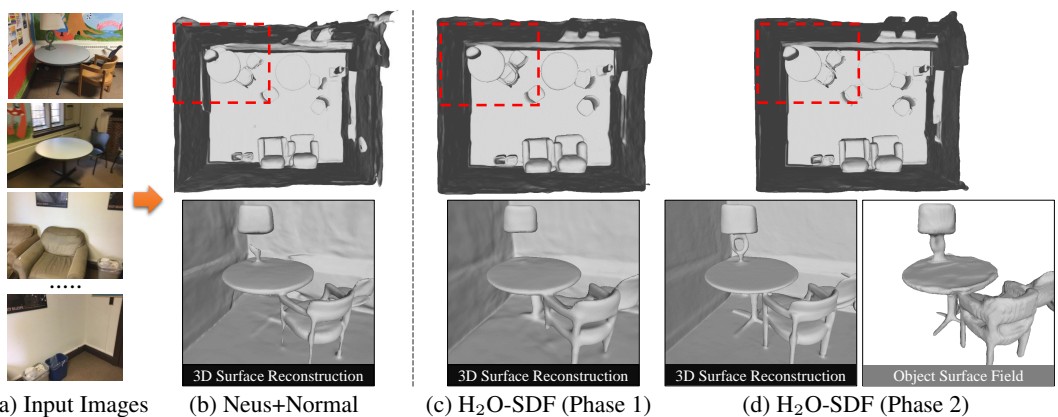

| (a) Input Images | (b) Neus+Normal | (c) H$_2$O-SDF (Phase 1) | (d) H$_2$O-SDF (Phase 2) |

Figure 1: Comparison of Reconstruction Results

Motivated by the previous limitations and the inherent complexity of reconstructing indoor environments, we introduce a two-phase learning approach called **H$_2$O-SDF** that stands for **H**olistic surface learning **to O**bject surface learning for **SDF**. The first phase, focuses on global indoor scene geometry, while the second phase zooms in on the intricate geometrical and surface details of individual objects within the indoor space. For the initial phase, holistic surface learning, we present a novel rendering loss re-weighting scheme based on normal uncertainty. This effectively addresses the issues of over-smoothing and discontinuity that arise due to conflicting information from surface normals and colors. In the second phase, object surface learning, dedicated to indoor object surface learning, we recognize that rendering-based consistency alone cannot fully recover fine-grained surface details of the given objects. To overcome this, we propose a new object surface representation called the Object Surface Field (OSF). OSF still leverages SDF to capture object geometries and surfaces but they complement each other in inducing the 3D geometry reconstruction without the need for direct SDF value supervision. It effectively solves the vanishing gradient problem. Furthermore, we enhance the performance of reconstruction by leveraging an advanced OSF-guided sampling technique. As a result, object surface learning achieve more fine-grained capturing of surface details of individual objects. Fig. 1 illustrates the superiority of our approach in comparison to other existing methods like NeuS with additional normal priors (Fig. 1(b) vs Fig. 1(c,d)). Through comprehensive experiments conducted on the ScanNet dataset (Dai et al., 2017), we establish the superior performance of our method over other state-of-the-art solutions.

In summary, our main contributions are as follows:

- We present a novel two-phase learning approach, which comprises holistic surface learning and object surface learning. This approach enables us to distinguish effectively between object and non-object regions in indoor scenes. Consequently, we preserve the overarching room layout while concurrently capturing intricate details within specific object areas.
- The core driver of object surface learning is a novel concept known as the Object Surface Field (OSF). This concept enhances previous SDF to extract watertight surfaces of objects in 3D space by overcoming the vanishing gradient issue of SDF. We propose a targeted sampling strategy for OSF that can further improve the quality of reconstruction, particularly in areas requiring high-frequency detail.
- Our methods substantially outperform previous approaches in the domain of indoor scene reconstruction. This superiority is empirically validated through extensive experimental studies, including a series of ablation tests that demonstrate the efficacy of each individual component of our two-phase learning approach.

## 2 RELATED WORK

### 2.1 NEURAL IMPLICIT SURFACE REPRESENTATION

Inspired by the success of NeRF (Mildenhall et al., 2020), a neural field approach that encodes the geometry and radiance information of 3D coordinates using MLP has become popular. Subsequent

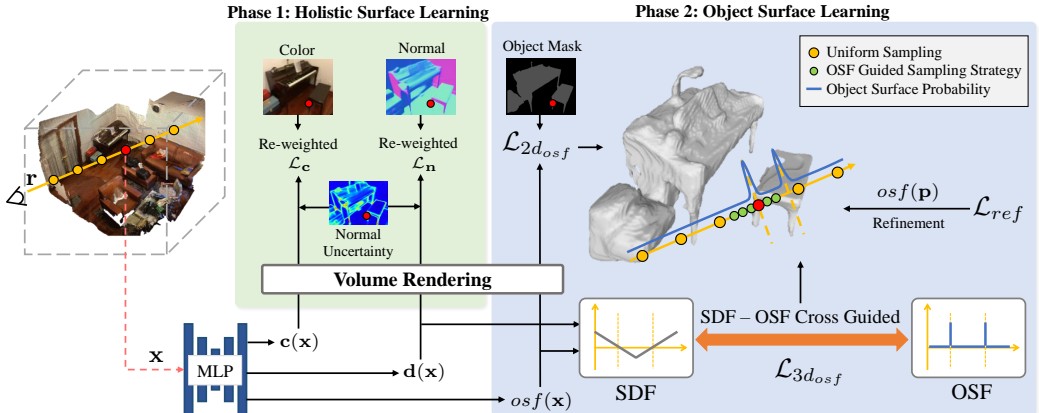

Figure 2: **Architecture Overview** The main pipeline consists of two phases. During the first phase (green part), we learn the global indoor scene geometry through re-weighted $\mathcal{L}_{\mathbf{c}}$ and $\mathcal{L}_{\mathbf{n}}$ based on normal uncertainty from an input position $\mathbf{x}$. During the second phase (blue part), we further train the Object Surface Field $osf(\mathbf{x})$ using $\mathcal{L}_{2d_{osf}}$ that is supervised by a 2D object mask; $\mathcal{L}_{3d_{osf}}$ that cross-guides between OSF $osf(\mathbf{x})$ and SDF $d(\mathbf{x})$; and $\mathcal{L}_{ref}$ that refines $osf$ with $\mathbf{p}$ (point cloud). During this process, we conduct OSF-guided sampling strategy (green dot).

works like NeuS (Wang et al., 2021) and VolSDF (Yariv et al., 2021) enable more accurate surface reconstruction by representing SDF as an implicit function. Based on these studies, recent works(Fu et al., 2022; Wang et al., 2022b) show that the inconsistency between the rendered color depth and the 0-value surface of the SDF could lead to less desirable surface reconstructions in practice. To address this issue, Geo-NeuS (Fu et al., 2022) proposes a method that aligns sparse point clouds extracted from SFM with the zero-level set of SDF. Furthermore, HF-NeuS (Wang et al., 2022b) concludes that it is challenging to represent both high-frequency and low-frequency components with a single SDF. To overcome such a challenge, a new methodology is proposed to decompose the SDF into their respective base functions and displacement functions. Although these methods improve reconstruction quality at the object level, they face challenges in reconstructing indoor scenes. The main reason is that indoor scenes include a significant proportion of low-frequency areas (e.g., walls, floors, ceilings), making it difficult to restore them using only color supervision.

## 2.2 NEURAL 3D RECONSTRUCTION FOR INDOOR SCENES

Recently, other research works focus on inducing smoothness in the reconstruction process to improve low-frequency surfaces by incorporating geometric cues, including semantic priors (Guo et al., 2022), normal priors (Wang et al., 2022a) and depth priors (Yu et al., 2022) that are provided by pre-trained models. Due to the inherent properties of the neural network, low-frequency surfaces tend to converge more rapidly than complex multiple-object surfaces characterized by high-frequency surfaces (Tancik et al., 2020). Moreover, the multiple object surfaces are hard to recover even during the advanced learning stage due to the vanishing gradient problem. To address this issue, $I^2$-SDF proposes a method that back projects 2D depth into a 3D point cloud to guide the SDF explicitly. HelixSurf (Liang et al., 2023) shows the inaccuracy of 3D priors and proposes a method to leverage the benefits of both explicit and implicit methods. In contrast, our proposed method, $H_2O$-SDF, captures all intricate geometrical and surface details of objects while maintaining the smoothness of room layouts by learning object surface field.

## 3 OUR METHOD

We provide an overview of our approach in Fig. 2. Our $H_2O$-SDF pipeline consists of two phases: the first phase, holistic surface learning (Sec. 3.1), concentrates on the global scene geometry, and the second phase, object surface learning (Sec. 3.2), delves into the intricate geometrical and surface details of objects within the indoor scene. This dual-phase learning approach achieves a nuanced balance, carefully preserving the geometric integrity of room layouts while also capturing intricate surface details of specific objects.

### 3.1 HOLISTIC SURFACE LEARNING

Holistic surface learning is dedicated to reconstructing the overall geometry, including the smooth room layout, which constitutes a significant portion of the indoor scene and predominantly features low-frequency regions. We represent the scene's geometry as the signed distance function (SDF) $d(\mathbf{x})$ for each spatial position $\mathbf{x}$. Practically, the SDF function $d$ is realized through a multi-layer perceptron (MLP). The geometry network is formulated as $f_g : \mathbf{x} \in \mathbb{R}^3 \mapsto (d(\mathbf{x}) \in \mathbb{R}, \mathbf{z}(\mathbf{x}) \in \mathbb{R}^{256})$, where $\mathbf{z}$ is a geometrical feature. The view-dependent color $\mathbf{c}$ is also implemented by an MLP. The color network is articulated as $f_c : (\mathbf{x} \in \mathbb{R}^3, \mathbf{v} \in \mathbb{R}^3, \mathbf{n}(\mathbf{x}) \in \mathbb{R}^3, \mathbf{z}(\mathbf{x}) \in \mathbb{R}^{256}) \mapsto \mathbf{c}(\mathbf{x}, \mathbf{v}) \in \mathbb{R}^3$, where $\mathbf{v}$ is the view direction and $\mathbf{n}(\mathbf{x}) = \nabla d(\mathbf{x})$ is the spatial gradient of SDF at point $\mathbf{x}$. According to the volume rendering formula, the colors along a ray $\mathbf{r}$ are rendered by $\hat{\mathbf{C}}(\mathbf{r}) = \sum_{i=1}^{N} T_i \cdot \alpha_i \cdot \mathbf{c}(\mathbf{x}_i, \mathbf{v}_i)$, where $N$ is the number of sampled points along the ray, $T_i = \prod_{j=1}^{i-1}(1 - \alpha_j)$ represents the accumulated transmittance, and $\alpha_i = 1 - \exp(-\int_{t_i}^{t_{i+1}} \rho(t)dt)$ is the discrete opacity, with the opaque density $\rho(t)$ adhering to the original definition in NeuS (Wang et al., 2021). Similarly, we render the surface normal along a ray by $\hat{\mathbf{n}}(\mathbf{r}) = \sum_{i=1}^{N} T_i \cdot \alpha_i \cdot \mathbf{n}(\mathbf{x}_i)$.

However, as highlighted by NeuRIS (Wang et al., 2022a), there exists an inherent conflict between color and normal information. Normal priors can offer precise geometric guidance in texture-less regions, but color images might provide misleading appearance supervision in those regions due to insufficient visual features. Conversely, in regions with fine details, they exhibit the opposite behavior. That is, if color and normal losses are uniformly applied to all pixels, it results in discontinuities in planar regions, and details are overly smoothed. To mitigate this situation, we introduce a technique to re-weight normal and color loss using normal uncertainty, which exhibits low uncertainty in planar regions and high uncertainty in texture-rich surfaces.

We initially estimate the normal $\hat{\mathbf{n}}(\mathbf{r})$ and the corresponding uncertainty $u_\mathbf{r}$ of each pixel using a pre-existing monocular normal estimation model (Bae et al., 2021). Subsequently, we determine the weight of normal loss $\lambda_\mathbf{n}$ and color loss $\lambda_\mathbf{c}$ with $(\beta_\mathbf{n} - u_\mathbf{r})$ and $(\beta_\mathbf{c} + u_\mathbf{r})$ respectively, where $\beta_\mathbf{n}$ and $\beta_\mathbf{c}$ are two trade-off hyperparameters and $u_\mathbf{r}$ is the normal uncertainty of the ray $\mathbf{r}$. Consequently, planar areas with low uncertainty, such as room layout and large objects, are predominantly influenced by normal rather than color loss, leading to more coherent and smoother planes. Conversely, texture-rich areas are more influenced by color, preserving details and maintaining sharpness. The normal and color loss are defined below with each corresponding ground truth $\mathbf{n}(\mathbf{r})$ and $\mathbf{C}(\mathbf{r})$.

$$\mathcal{L}_\mathbf{n} = \sum_{\mathbf{r} \in \mathcal{R}} \|\hat{\mathbf{n}}(\mathbf{r}) - \mathbf{n}(\mathbf{r})\|_1 \cdot (\beta_\mathbf{n} - u_\mathbf{r}), \quad \mathcal{L}_\mathbf{c} = \sum_{\mathbf{r} \in \mathcal{R}} \left\|\hat{\mathbf{C}}(\mathbf{r}) - \mathbf{C}(\mathbf{r})\right\|_1 \cdot (\beta_\mathbf{c} + u_\mathbf{r}) \quad (1)$$

We employ the Eikonal loss (Gropp et al., 2020) to further regularize the gradients of SDF: $\mathcal{L}_{eik} = \sum_{i=1}^{N} \frac{1}{N}(\|\nabla d(\mathbf{x}_i)\|_2 - 1)^2$

### 3.2 OBJECT SURFACE LEARNING

Object surface learning is propelled by a novel concept that we introduce as object surface field (OSF). OSF directs SDF to encapsulate small-scale geometry and high-frequency details on objects while preserving the smoothness of room layout. The 2D object surface loss is a preliminary step in obtaining the initial value of OSF. The 3D object surface loss, enables OSF to carefully learn the object surface from SDF, letting SDF to capture high-frequency details under the guidance of OSF. To further improve the reconstruction performance, we leverage OSF-guided sampling strategy to prioritize object surfaces.

**2D Object Surface Loss** Object surface field (OSF) essentially represents the probability of object surface $osf(\mathbf{x})$ for each spatial point $\mathbf{x}$. It is realized by an MLP $f_o : (\mathbf{x} \in \mathbb{R}^3, \mathbf{z}(\mathbf{x}) \in \mathbb{R}^{256}) \mapsto osf(\mathbf{x}) \in \mathbb{R}$, and the OSF of the ray $\mathbf{r}$ is rendered by: $osf(\mathbf{r}) = \sum_{i=1}^{N} T_i \cdot \alpha_i \cdot osf(\mathbf{x}_i)$. As a preliminary step in learning OSF, we define 2D object surface loss, $\mathcal{L}_{2d_{osf}} = BCE(osf(\mathbf{r}), \mathbb{1}_o(\mathbf{r}))$

where $BCE$ is the binary cross-entropy loss, and the indicator function $\mathbb{1}_o$ returns 1 if the object surface exists along the ray, and 0 otherwise. It is determined by the 2D object mask, which can be easily obtained using a pre-trained model (Lambert et al., 2020). $\mathcal{L}_{2d_{osf}}$ is utilized to minimize the discrepancy between the predicted OSF of the ray and the given 2D object mask, thereby establishing the initial value of OSF.

**3D Object Surface Loss** 3D object surface loss encourages SDF to capture the high-frequency details of objects by interacting with OSF. As depicted in Fig. 3(a), while $\mathcal{L}_{2d_{osf}}$ trains OSF to learn the approximate object region, there is still a lack of information from the 2D object mask for OSF to learn the precise object boundary. Also, as it employs volume rendering to accumulate $osf$ based on density-based weight, it struggles to provide supervision to empty spaces with low density, leading to a noisy OSF. Conversely, as Fig. 3(b) illustrates, the 3D object surface loss enables OSF to learn the object surface meticulously, allowing it to provide a supervision signal to the object surface of SDF. For this, we propose the 3D object surface loss as below:

$$\mathcal{L}_{3d_{osf}} = \frac{1}{N}\sum_{i=1}^{N}\left[\mathbb{1}_o(\mathbf{r})\cdot osf(\mathbf{x}_i)\cdot|osf(\mathbf{x}_i)-\sigma_\gamma(\mathbf{x}_i)| + (1-\mathbb{1}_o(\mathbf{r}))\cdot osf(\mathbf{x}_i)\right] \quad (2)$$

where, the scaled sigmoid function $\sigma_\gamma(\mathbf{x}) = \frac{1}{1+\exp(\gamma\cdot d(\mathbf{x}))}$, $\gamma$ is a hyperparameter determining the steepness of the function, and $N$ is the number of sampled points along the ray $\mathbf{r}$. This loss functions differently depending on the existence of the object surface along the ray.

The 3D object surface loss is designed for OSF to adhere to the object surface of SDF, simultaneously learning low probability on the non-object surfaces effectively. When the ray intersects with the object surface, the probability of the object surface should alter according to the sign of the SDF. In essence, during the traversal of the ray within the zero-level set of the SDF, it is anticipated that the object surface probability should escalate upon entry and diminish upon exit. It is motivated by ObjectSDF (Wu et al., 2022), which explains the correlation between the SDF and the 3D semantic field using a scaled sigmoid function. Unlike ObjectSDF, where SDF values are directly transformed into semantic field values and learned by 2D rendering-based loss, our method introduces a distinct object surface field to mutually guide the object surface of the SDF.

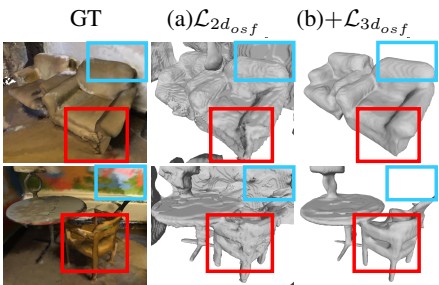

Figure 3: **Comparisons of OSF** Compared to using only (a) $\mathcal{L}_{2d_{osf}}$, introduction of our (b) $\mathcal{L}_{3d_{osf}}$ enables OSF to represent precise object boundaries. Improvement includes object surfaces (Red) and non-object surfaces (Blue).

When the ray intersects only with the room layout surface, the OSF of all points along the ray should exhibit low probability. Thus, $\mathcal{L}_{3d_{osf}}$ is structured to prevent OSF activation on the room-layout surface. In multi-view settings, it can effectively prompt OSF to decrease for occluded space located behind objects as well as the room layout. When the ray intersects with the room layout surface after colliding with the object surface, as the probability of OSF has already reduced, OSF remains unaffected by changes in SDF.

Moreover, we observe that the reliability of point clouds extracted from multi-view stereo (MVS) is superior in object areas with abundant visual features, while it is inferior in texture-less areas. As a consequence, we propose a refinement loss to aid the learning of the OSF with the point clouds. The refinement loss is defined as follows:

$$\mathcal{L}_{ref} = -\frac{1}{N_i}\sum_{\mathbf{x}_j\in\mathcal{P}_i}\mathbb{1}_o(\mathbf{x}_j)\cdot\log(osf(\mathbf{x}_j)) \quad (3)$$

Here, $N_i$ is the number of points in the point clouds $\mathcal{P}_i$ from view $V_i$, and $\mathbb{1}_o(\mathbf{x}_j)$ denotes whether an input point belongs to an object surface or not, returning a value of 1 or 0 respectively. This approach enhances the performance of the OSF network, allowing for more precise learning of intricate object regions with thin structures and high occlusion.

**Mutual Induction of OSF and SDF** In this section, we explain in more detail 1) how SDF directs OSF to learn the surface of the object and 2) how a well-learned OSF enables SDF to capture high-frequency geometric detail. As depicted in Fig. 4(a), during the initial stage of object surface learning, the OSF in regions with high rendering weight ($w = T\cdot\alpha$) are trained to converge towards 1 due to $\mathcal{L}_{2d_{osf}}$ (Eq. 3.2). Initially, the OSF represents high values not only on the surface but also inside the object. However, $\partial\mathcal{L}_{3d_{osf}}/\partial osf(\mathbf{x})$ is computed as follows:

$$\frac{\partial\mathcal{L}_{3d_{osf}}}{\partial osf(\mathbf{x})} = \begin{cases} \sigma_\gamma(\mathbf{x}) - 2\cdot osf(\mathbf{x}), & \text{if } osf(\mathbf{x}) < \sigma_\gamma(\mathbf{x}) \\ 2\cdot osf(\mathbf{x}) - \sigma_\gamma(\mathbf{x}), & \text{otherwise} \end{cases} \quad (4)$$

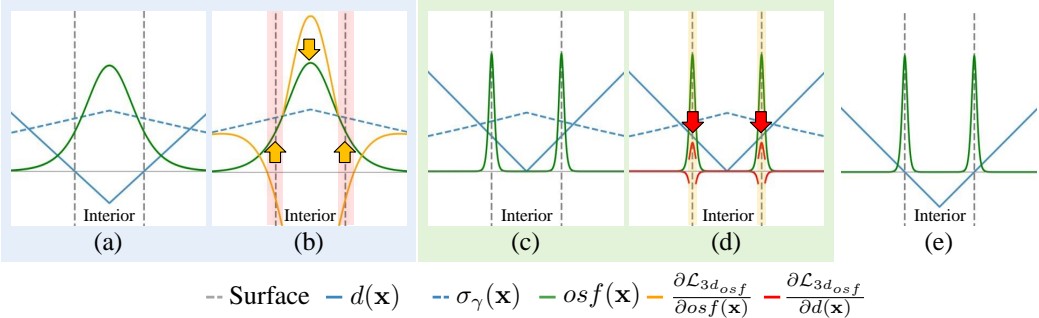

$$-- \text{Surface} \quad - d(\mathbf{x}) \quad -- \sigma_\gamma(\mathbf{x}) \quad - osf(\mathbf{x}) - \frac{\partial \mathcal{L}_{3d_{osf}}}{\partial osf(\mathbf{x})} - \frac{\partial \mathcal{L}_{3d_{osf}}}{\partial d(\mathbf{x})}$$

Figure 4: **Interaction of OSF and SDF** Illustration of (a) the initial status of OSF, (b) the influence of the gradient of $\mathcal{L}_{3d_{osf}}$ with respect to OSF, (c) the case when SDF fails to capture thin structure, (d) the influence of the gradient of $\mathcal{L}_{3d_{osf}}$ with respect to SDF, and (e) the final result of OSF and SDF. Interior refers to the region inside an object.

As illustrated in Fig. 4(b), $\partial \mathcal{L}_{3d_{osf}}/\partial osf(\mathbf{x})$ yields negative values only in the red region, when $osf(\mathbf{x}) < \sigma_\gamma(\mathbf{x}) < 2 \cdot osf(\mathbf{x})$. In other words, the OSF is induced to have high values in the region where the SDF indicates that surfaces are nearby, but the OSF has not yet learned about those surfaces. Through the learning process, the OSF is trained to decrease both within and outside the object while it increases only near the surface. Consequently, the OSF is guided to represent high values exclusively on the object surfaces. The following process describes how the well-trained OSF encourages the SDF to learn high-frequency details that have not yet been captured (see Fig. 4(c)). Considering the case that a ray crosses a thin structure such as a chair leg, $\partial \mathcal{L}_{3d_{osf}}/\partial d(\mathbf{x})$ yields the following results:

$$\frac{\partial \mathcal{L}_{3d_{osf}}}{\partial d(\mathbf{x})} = \begin{cases} \gamma \cdot osf(\mathbf{x}) \cdot \sigma_\gamma(\mathbf{x}) \cdot (1 - \sigma_\gamma(\mathbf{x})), & \text{if } \sigma_\gamma(\mathbf{x}) < osf(\mathbf{x}) \\ -\gamma \cdot osf(\mathbf{x}) \cdot \sigma_\gamma(\mathbf{x}) \cdot (1 - \sigma_\gamma(\mathbf{x})), & \text{otherwise} \end{cases} \tag{5}$$

As illustrated in Fig. 4(d), the SDF in regions of high OSF are pulled towards the negative side when $\sigma_\gamma(\mathbf{x}) < osf(\mathbf{x})$. This loss generates gradients to the thin structure, effectively addressing the vanishing gradient problem. By means of the Eikonal loss, the SDF can be trained to learn high-frequency details while maintaining gradients and a water-tight shape. Please refer to our appendix (Sec. A.2) for the derivation of the vanishing gradient problem in our base model.

**Object Surface Field Guided Sampling Strategy** In other approaches such as NeuS, hierarchical sampling relies on the volume rendering weight ($w = T \cdot \alpha$), derived from density. However, in typical indoor scenes, expansive planar areas, particularly room layouts, tend to stabilize early in the training phase, leaving high-frequency details insufficiently captured. Consequently, density-based sampling over-emphasizes already well-reconstructed room layout regions, ignoring points in high-frequency areas. The application of object surface field (OSF) has mitigated the overlooked density issues in high-frequency regions. There remains a necessity to prioritize object surfaces exhibiting more intricate structures compared to room layouts. To overcome this, we introduce a novel sampling strategy, guided by the OSF. After uniformly sampling points along the ray, we iteratively perform importance sampling on the probability distribution calculated from $w(\mathbf{x}) \cdot osf(\mathbf{x})$. This approach lets the network to concentrate more on areas with high density and object surface probability. Consequently, the OSF-guided sampling strategy enhances the overall model to more precisely capture high-frequency details. Details of the OSF-guided sampling strategy are provided in the Appendix (Sec. A.2).

## 4 EXPERIMENTS

### 4.1 EXPERIMENTAL SETTING

**Datasets** To evaluate the effectiveness of our proposed algorithm in real-world scenarios, we conduct empirical analysis using ScanNet (Dai et al., 2017). ScanNet is a large-scale dataset, comprising 1613 scenes with 2.5 million views. Each scene contains a sequence of frames annotated with calibrated intrinsic and extrinsic camera parameters, in addition to surface reconstruction. We choose 4 scenes from (Guo et al., 2022) and randomly selected additional 8 scenes.

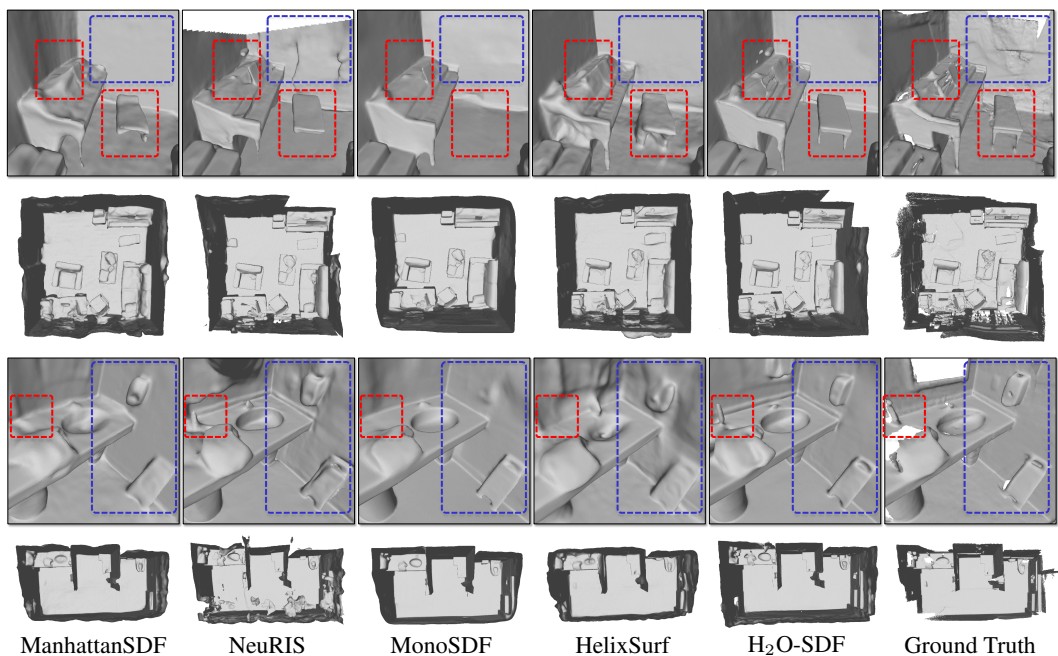

| ManhattanSDF | NeuRIS | MonoSDF | HelixSurf | H$_2$O-SDF | Ground Truth |

Figure 5: **3D Reconstruction Results on ScanNet** H$_2$O-SDF shows improved reconstruction ability for both room-layout regions (blue box) and fine-grained object regions (red box) compared to other methods. Reconstruction for the remaining scenes are visualized in the Appendix (Sec. A.3)

**Implementation Details** We use the NeuS (Wang et al., 2021) as our base model, which comprises a geometry network with an 8-layer MLP and an appearance network with a 4-layer MLP. To learn the object surface field, we employ an object surface network with a 4-layer MLP. For model optimization, we adopt the Adam optimizer with an initial learning rate of 2e-4 and then train H$_2$O-SDF with batches of 512 rays. We conducted 40k iterations of training during the holistic surface learning phase, followed by an additional 120k iterations during the object surface learning phase. We use the following hyperparameters in our experiments: $\beta_\mathbf{c} = 1.0$, $\beta_\mathbf{n} = 2.0$, $\lambda_{eik} = 0.1$, $\lambda_{2d_{osf}} = 0.5$, $\lambda_{3d_{osf}} = 0.5$, $\lambda_{ref} = 0.1$. Our code is implemented with PyTorch (Paszke et al., 2017), conducting all experiments on a single NVIDIA RTX 3090Ti GPU. We provide more implementation details in the Appendix (Sec. A.1).

**Compared Methods** We compare our H$_2$O-SDF against other traditional MVS method such as COLMAP (Schönberger et al., 2016) and neural volume rendering methods, including NeRF (Mildenhall et al., 2020), VolSDF (Yariv et al., 2021), NeuS (Wang et al., 2021), ManhattanSDF (Guo et al., 2022), NeuRIS (Wang et al., 2022a), MonoSDF (Yu et al., 2022), and HelixSurf (Liang et al., 2023).

**Metrics** For a quantitative comparison of 3D surface reconstruction, we utilize five standard metrics introduced in (Murez et al., 2020): Accuracy, Completeness, Precision, Recall, and F-score. Following (Sun et al., 2021), we adopt F-score as the comprehensive metric to measure the quality of 3D reconstruction, as it considers both accuracy and completeness.

## 4.2 COMPARISONS

**3D Reconstruction** To demonstrate the reconstruction ability of H$_2$O-SDF, we provide qualitative and quantitative comparisons with other state-of-the-art methods. Our method demonstrates significantly smoother reconstruction results in low-frequency areas compared to other methods, highlighting the benefits of our re-weighting scheme (See Fig. 5). Moreover, thanks to the object surface learning, H$_2$O-SDF can better represent the fine-grained surface details of object geometries (e.g., chair legs or lamp) compared to other methods. Additionally, as shown in Tab. 1, our method outperforms previous methods, achieving a new state-of-the-art performance in every 3D geometry metric. These results, altogether, demonstrate the effectiveness of our two-phased learning approach by substantially improving the overall quality of the indoor scene reconstruction.

| Methods | Acc.↓ | Comp.↓ | Prec.↑ | Recall↑ | F-score↑ |
|---|---|---|---|---|---|
| COLMAP | 0.047 | 0.235 | 0.711 | 0.441 | 0.537 |
| ACMP | 0.118 | 0.081 | 0.531 | 0.581 | 0.555 |
| NeRF | 0.735 | 0.177 | 0.131 | 0.290 | 0.176 |
| VolSDF | 0.414 | 0.120 | 0.321 | 0.394 | 0.346 |
| NeuS | 0.179 | 0.208 | 0.313 | 0.275 | 0.291 |
| ManhattanSDF | 0.053 | 0.056 | 0.715 | 0.664 | 0.688 |
| NeuRIS | 0.052 | 0.050 | 0.713 | 0.677 | 0.690 |
| MonoSDF | 0.035 | 0.048 | 0.799 | 0.681 | 0.733 |
| HelixSurf | 0.038 | 0.044 | 0.786 | 0.727 | 0.755 |
| **$H_2$O-SDF** | **0.032** | **0.0373** | **0.834** | **0.769** | **0.799** |

Table 1: **Quantitative Comparison on ScanNet** We compare $H_2$O-SDF against other previous state-of-the-art methods.

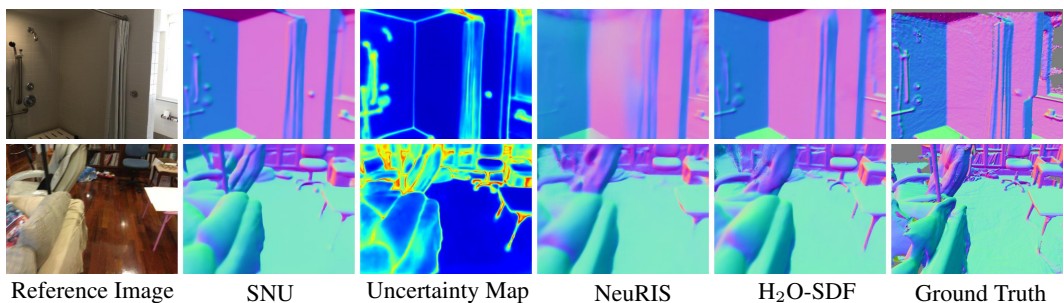

Reference Image     SNU     Uncertainty Map     NeuRIS     $H_2$O-SDF     Ground Truth

Figure 6: **Qualitative Normal Comparisons on ScanNet** Normal prior (SNU) and uncertainty map obtained from (Bae et al., 2021) is utilized in our re-weight scheme.

**Normal Predictions** We compare the normal prediction quality of $H_2$O-SDF against a monocular normal estimation method in (Bae et al., 2021) and other normal prior based reconstruction method in Wang et al. (2022a). As illustrated in Fig. 6, $H_2$O-SDF excels at reconstructing both broad, low-frequency regions and fine-grained surface details compared to other methods. More normal comparison results are provided in the Appendix (Sec. A.3).

**Generalizability** To further investigate the applicability and generalization of $H_2$O-SDF, we conducted additional experiments on the Replica (Straub et al., 2019) dataset and 7-Scenes (Shotton et al., 2013) dataset. As shown in Tab. 2, our method achieves superior 3D reconstruction results, irrespective of any specific domain of application.

| Model | Accu.↓ | Comp.↓ | Prec.↑ | Recall↑ | F-score↑ | Model | Accu.↓ | Comp.↓ | Prec.↑ | Recall↑ | F-score↑ |
|---|---|---|---|---|---|---|---|---|---|---|---|
| ManhattanSDF | **0.112** | 0.132 | 0.351 | 0.326 | 0.336 | ManhattanSDF | 0.131 | 0.417 | 0.644 | 0.407 | 0.492 |
| NeuRIS | 0.133 | 0.132 | 0.405 | 0.424 | 0.410 | NeuRIS | 0.023 | 0.027 | 0.921 | 0.898 | 0.909 |
| MonoSDF | 0.158 | 0.328 | 0.286 | 0.262 | 0.301 | MonoSDF | 0.020 | 0.017 | 0.951 | 0.923 | 0.934 |
| $H_2$O-SDF | 0.128 | **0.129** | **0.416** | **0.469** | **0.447** | $H_2$O-SDF | **0.016** | **0.025** | **0.983** | **0.935** | **0.957** |
| (a) Comparison results for 7-Scenes | | | | | | (b) Comparison results for Replica | | | | | |

Table 2: Quantitative Comparsion on 7-Scenes and Replica datasets

### 4.3 ANALYSIS

**Ablation Study** In this section, to validate the effectiveness of individual components in $H_2$O-SDF, we conduct ablation studies in different settings: (1) **NeuS**: Base model, (2) **Model A**: NeuS with normal priors, (3) **Model B**: the model undergoes training solely through our first stage, Holistic Surface Learning defined in Sec. 3.1, (4) **Model C**: Model B with Object Surface Field (OSF) defined in Sec. 3.2. Fig. 7 illustrates how the proposed components enhance reconstruction performance. Model-A demonstrates the ability to reconstruct low-frequency areas but still exhibits noise. With the proposed HSL, Model-B yields smoother and more consistent room layout surfaces than Model-A. By incorporating the OSF, we observe that it better captures intricate details within object regions, such as stands and chair legs. Finally, incorporating with OGS, our full model more suc-

| Method | NeuS | Normal | Reweight | OSF | OGS | Acc↓ | Comp↓ | Prec↑ | Recall↑ | F-score↑ |
|---|---|---|---|---|---|---|---|---|---|---|
| NeuS | ✓ | - | - | - | - | 0.132 | 0.114 | 0.376 | 0.334 | 0.353 |
| Model A | ✓ | ✓ | - | - | - | 0.040 | 0.042 | 0.794 | 0.742 | 0.766 |
| Model B | ✓ | ✓ | ✓ | - | - | 0.041 | 0.041 | 0.802 | 0.751 | 0.776 |
| Model C | ✓ | ✓ | ✓ | ✓ | - | 0.039 | 0.041 | 0.817 | 0.762 | 0.787 |
| $H_2O$-SDF | ✓ | ✓ | ✓ | ✓ | ✓ | **0.037** | **0.039** | **0.830** | **0.773** | **0.800** |

Table 3: Quantitative Results on the Ablation Study of 12 scene ScanNet. Reweight means re-weighting scheme for Holistic Surface Learning (Sec. 3.1). OSF indicates $\mathcal{L}_{2d_{osf}} + \mathcal{L}_{3d_{osf}} + \mathcal{L}_{ref}$ for Object Surface Field (Sec. 3.2). OGS means OSF-Guided Sampling strategy

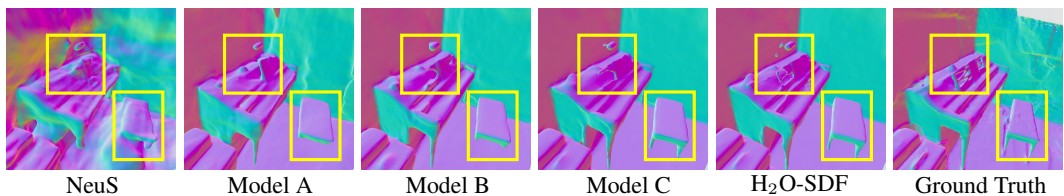

|   NeuS   |   Model A   |   Model B   |   Model C   |   $H_2O$-SDF   |   Ground Truth   |

Figure 7: Visualization Results of our Ablation Study

cessfully facilitates the reconstruction of fine-grained detailed regions. Quantitative results in Tab. 3 demonstrate the effectiveness of $H_2O$-SDF through the progressive addition of different components to the baseline model. More ablation studies are provided in the Appendix (Sec. A.3).

**Object Surface Learning** As shown in Fig. 8, we conduct additional analysis on the effectiveness of object surface learning in the case of thin structure regions. NeuS with a normal prior exhibit higher weights in the floor regions characterized by low-frequency surfaces rather than in thin structure regions due to the vanishing gradient problem. On the other hand, the introduction of object surface field (OSF) exhibits peak values in the thin structure regions with zero SDF values, as illustrated in Fig. 8(c). This demonstrates that OSF directs gradients to thin structures. By incorporating the OSF-Guided Sampling strategy (OGS), $H_2O$-SDF resolves weight ambiguity more effectively.

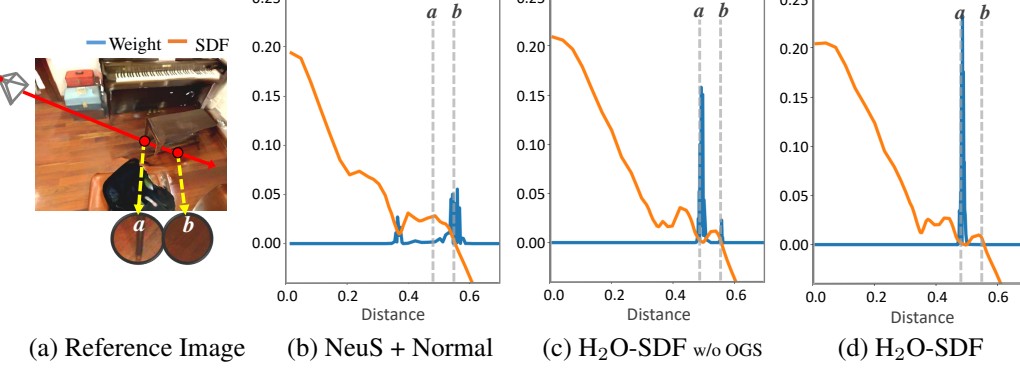

(a) Reference Image     (b) NeuS + Normal     (c) $H_2O$-SDF w/o OGS     (d) $H_2O$-SDF

Figure 8: **Analysis on Object Surface Learning** For visualization, we extract the weights distribution and SDF values from a set of points along the emitted ray corresponding to a single pixel.

## 5 CONCLUSION

We introduce a novel two-phase learning approach for 3D indoor reconstruction. The first phase, holistic surface learning, reconstructs the overall geometry of a scene, resolving conflicts in color and normal information. The second phase, object surface learning, addresses the vanishing gradient problem and enables the capturing of detailed object surface geometry by leveraging the object surface field (OSF). This concept is interesting as OSF extends beyond existing 2D priors, functioning as a 3D geometry cue that delivers 3D spatial information to the SDF. Our method has proven to achieve state-of-the-art reconstruction accuracy and smoothness on ScanNet. Future work will focus on integrating faster convergence radiance fields (e.g. TensoRF (Chen et al., 2022)) to expedite training and developing applications like scene-editing using OSF.

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

# A APPENDIX

## A.1 ADDITIONAL IMPLEMENTATION DETAILS

**Network Architecture** Our detailed network architecture is illustrated in Fig. 9. Our method uses three networks to encode the implicit radiance field, implicit signed distance field, and object surface field. The geometry network takes as input position $\mathbf{x}$ mapped by positional encoding (Mildenhall et al., 2020) and outputs SDF values and geometry features.

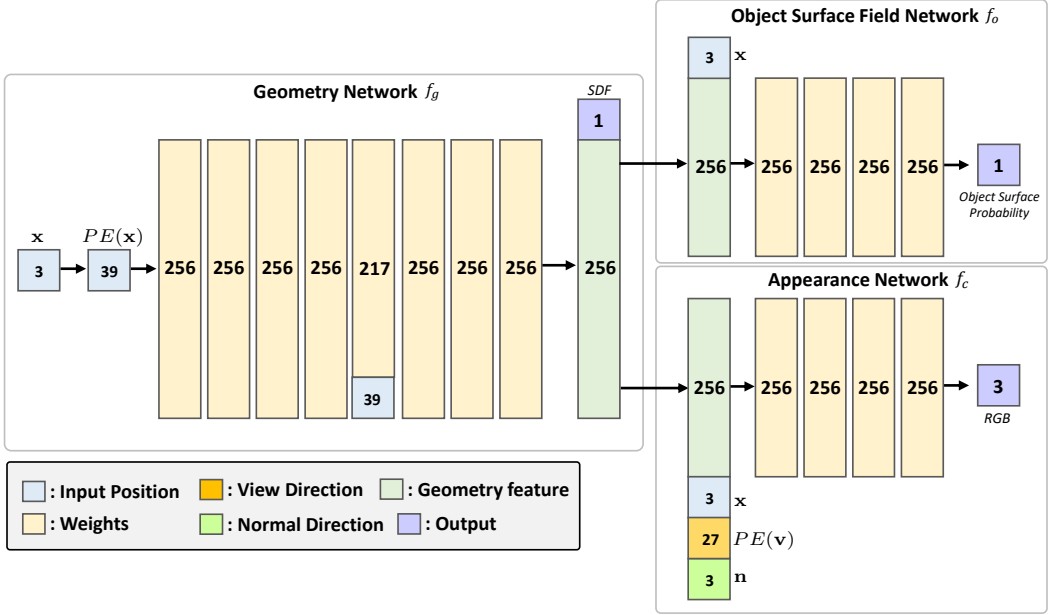

Figure 9: **Network Architecture** Our network takes as input position $\mathbf{x}$ and the view direction $\mathbf{v}$ of the sample point, and outputs SDF, color and probability of object surface. Additionally, the input position $\mathbf{x}$ and view direction $\mathbf{v}$ are mapped by a positional encoding $PE(\cdot)$.

**Experimental Setting** We test our method with 4 scenes as in the previous study (Wei et al., 2021; Wang et al., 2022a; Liang et al., 2023) and an additional 8 scenes, which were randomly selected. For each scene, we uniformly sample frames at a rate of one-tenth or one-sixth proportional to the video length. These frames are resized in $640 \times 480$ resolution. We adopted SNU (Bae et al., 2021) as our normal estimation module, which takes a single image as input and produces a predicted normal map and uncertainty map as output. To facilitate the re-weighting scheme, we normalized the uncertainty map to a range between 0 and 1. Additionally, we take MSeg (Lambert et al., 2020) as our semantic segmentation estimation module, employing its officially provided pre-trained model. To create the object mask used in object surface learning, we masked the predicted segmentation results by assigning a value of 1 to categories related to objects and 0 to all other categories.

**Evaluation Metrics** To evaluate the 3D reconstruction surface quality, we use the Accuracy, Completeness, Precision, Recall, and F-score with a threshold of 5cm. The definitions of 3D reconstruction metrics are shown in Tab. 4(a). $\mathcal{P}$ and $\mathcal{P}^*$ are the set of points sampled from the predicted mesh and ground truth mesh. We use four standard metrics to evaluate the rendered normal quality, and definitions of normal evaluation metrics are shown in Tab. 4(b). $\mathbf{n}$ and $\mathbf{n}^*$ are the predicted normal vector and ground truth normal vector.

## A.2 ADDITIONAL METHOD DETAILS

**Training step** We train $H_2O$-SDF in two phases: *Phase 1* - a holistic surface learning step, incorporating $\mathcal{L}_{hol} = \lambda_{\mathbf{c}} \cdot \mathcal{L}_{\mathbf{c}} + \lambda_{\mathbf{n}} \cdot \mathcal{L}_{\mathbf{n}} + \lambda_{eik} \cdot \mathcal{L}_{eik}$, and *Phase 2* - an object surface learning step, adding $\mathcal{L}_{obj} = \lambda_{2d_{osf}} \cdot \mathcal{L}_{2d_{osf}} + \lambda_{3d_{osf}} \cdot \mathcal{L}_{3d_{osf}} + \lambda_{ref} \cdot \mathcal{L}_{ref}$, with an OSF-guided sampling in the later stage. The final loss function is $\mathcal{L}_{hol} + \mathcal{L}_{obj}$.

**Vanishing Gradient Problem** In this section, we explain the vanishing gradient problem that exists in SDF-based neural representation. The explanation is motivated by I$^2$-SDF (Zhu et al., 2023) which is based on VolSDF (Yariv et al., 2021). Unlike the paper, our model is based on NeuS (Wang et al., 2021), so we demonstrate the gradient vanishing problem on the NeuS based method.

Suppose the loss for the view-dependent color $\mathbf{c}$ is a function $\mathcal{L} = \mathbf{c}(\rho(d(\mathbf{x})))$, where $\rho$ is the density and $d(\mathbf{x})$ is a signed distance function (SDF) for point $\mathbf{x}$. The derivative of $\mathcal{L}$ with respect to $\mathbf{x}$ can be written as $\frac{\partial \mathcal{L}}{\partial \mathbf{c}} \frac{\partial \mathbf{c}}{\partial \rho} \frac{\partial \rho}{\partial d} \frac{\partial d}{\partial \mathbf{x}}$ (Zhu et al., 2023). The following derivation is to show the derivative of $\mathcal{L}$ vanishes rapidly as point $\mathbf{x}$ is getting far from the surface.

According to NeuS, for $\rho(\mathbf{x}) > 0$, density is defined as

$$\rho(\mathbf{x}) = \frac{-\frac{\mathrm{d}\Phi_s}{\mathrm{d}\mathbf{x}}(d(\mathbf{x}))}{\Phi_s(d(\mathbf{x}))} \tag{6}$$

where

$$\Phi_s(d(\mathbf{x}))) = \frac{1}{1 + e^{-s \cdot d(\mathbf{x})}} \tag{7}$$

$\frac{\mathrm{d}\Phi_s}{\mathrm{d}\mathbf{x}}(d(\mathbf{x}))$ can be computed by

$$\frac{\partial \Phi_s}{\partial d} \frac{\partial d}{\partial \mathbf{x}} = \frac{se^{-s \cdot d(\mathbf{x})}}{(e^{-s \cdot d(\mathbf{x})} + 1)^2} \cdot \frac{\partial d}{\partial \mathbf{x}} \tag{8}$$

Note that $d$ follows the characteristic of SDF by Eikonal loss that,

$$\frac{\partial d}{\partial \mathbf{x}} \approx 1 \tag{9}$$

Hence,

$$\frac{\mathrm{d}\Phi_s}{\mathrm{d}\mathbf{x}}(d(\mathbf{x})) = \frac{se^{-s \cdot d(\mathbf{x})}}{(e^{-s \cdot d(\mathbf{x})} + 1)^2} \tag{10}$$

Therefore, if we express $\rho(\mathbf{x})$ in terms of $d(\mathbf{x})$,

$$\rho(\mathbf{x}) = \frac{-\frac{se^{-s \cdot d(\mathbf{x})}}{(e^{-s \cdot d(\mathbf{x})} + 1)^2}}{\frac{1}{1 + e^{-s \cdot d(\mathbf{x})}}} = -\frac{se^{-s \cdot d(\mathbf{x})}}{1 + e^{-s \cdot d(\mathbf{x})}} \tag{11}$$

In this case, the partial derivative of $\rho$ with respect to $d$ can be calculated as:

$$\frac{\partial \rho}{\partial d} = \frac{s^2 \cdot e^{s \cdot d(\mathbf{x})}}{(e^{s \cdot d(\mathbf{x})} + 1)^2} \tag{12}$$

Standard deviation $1/s$ is a trainable parameter that approaches to zero as the network training converges (Wang et al., 2021). In other words, $s$ is trained with a sufficiently large value.

| Methods | Definition | Methods | Definition |
|---|---|---|---|
| Accuracy | $\mathrm{mean}_{\mathbf{p} \in \mathcal{P}}(\min_{\mathbf{p}^* \in \mathcal{P}^*} \|\mathbf{p} - \mathbf{p}^*\|)$ | Mean | $\frac{1}{N} \sum \cos^{-1}\left[\frac{|\mathbf{n} \cdot \mathbf{n}^*|}{|\mathbf{n}||\mathbf{n}^*|}\right]$ |
| Completeness | $\mathrm{mean}_{\mathbf{p}^* \in \mathcal{P}^*}(\min_{\mathbf{p} \in \mathcal{P}} \|\mathbf{p} - \mathbf{p}^*\|)$ | Median | $\mathrm{median}\left\{\cos^{-1}\left[\frac{|\mathbf{n} \cdot \mathbf{n}^*|}{|\mathbf{n}||\mathbf{n}^*|}\right]\right\}$ |
| Precision | $\mathrm{mean}_{\mathbf{p} \in \mathcal{P}}(\min_{\mathbf{p}^* \in \mathcal{P}^*} \|\mathbf{p} - \mathbf{p}^*\| < 0.05)$ | RMSE | $\sqrt{\frac{1}{N} \sum (\cos^{-1}\left[\frac{|\mathbf{n} \cdot \mathbf{n}^*|}{|\mathbf{n}||\mathbf{n}^*|}\right])^2}$ |
| Recall | $\mathrm{mean}_{\mathbf{p}^* \in \mathcal{P}^*}(\min_{\mathbf{p} \in \mathcal{P}} \|\mathbf{p} - \mathbf{p}^*\| < 0.05)$ | **deg**$^\circ$ | $\frac{1}{N}\{\mathbf{n}, \mathbf{n}^* : cos^{-1}\left[\frac{|\mathbf{n} \cdot \mathbf{n}^*|}{|\mathbf{n}||\mathbf{n}^*|}\right] < \mathbf{deg}^\circ\}$ |
| F-score | $\frac{2 \times \mathrm{Precision} \times \mathrm{Recall}}{\mathrm{Precision} + \mathrm{Recall}}$ | | |

(a) Evaluation Metrics for 3D Reconstruction Result      (b) Evaluation Metrics for Rendered Normal Result

Table 4: Evaluation Metrics

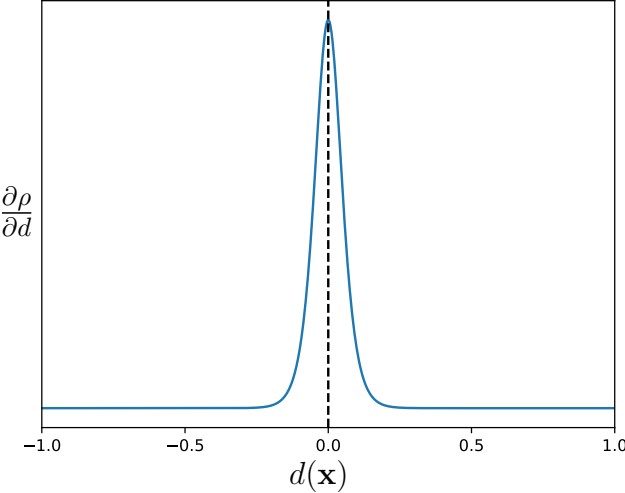

Figure 10: Graph of $\partial\rho/\partial d$ with respect to $d(\mathbf{x})$

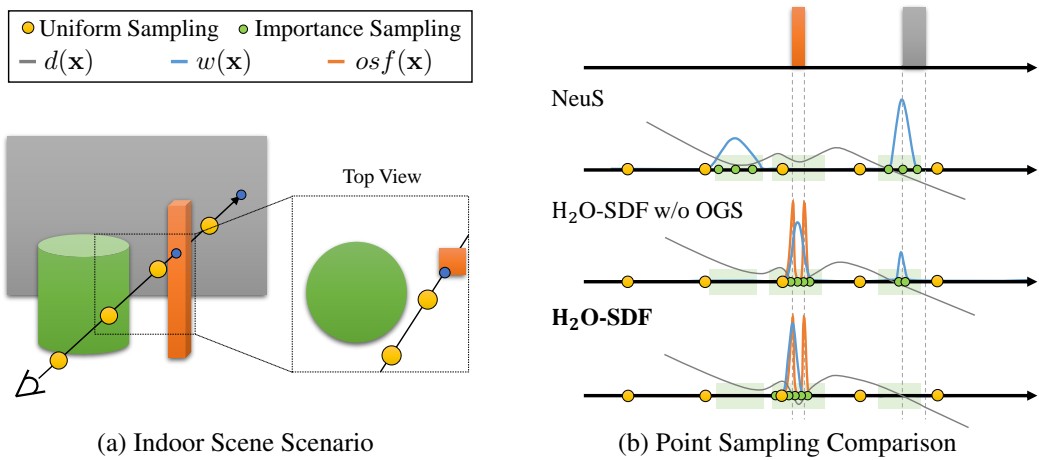

(a) Indoor Scene Scenario                (b) Point Sampling Comparison

Figure 11: Illustration of (a) simplified indoor scene scenario, and (b) comparison of point sampling schemes between the previous method and ours. OGS represents OSF-guided sampling strategy.

The graph of $\partial\rho/\partial d$ is shown in Fig. 10, and as $s$ increases, the width of the peak in the graph narrows. As a result, as $d(\mathbf{x})$ gets larger, which means the point $\mathbf{x}$ becomes far from the surface, $\partial\rho/\partial d$ becomes near zero rapidly. It indicates the gradient of loss vanishes fast with increasing $d(\mathbf{x})$.

**Object Surface Field Guided Sampling Strategy Details** In this section, we describe in detail about object surface field guided point sampling strategy with an example. Fig. 11(a) shows a simplified indoor scene scenario in which multiple objects and room layout exist. In this case, a ray passes by a green object while hitting an orange object and a room layout (gray).

In the previous methods like NeuS, importance sampling is performed based on volume rendering weight ($w = T \cdot \alpha$) that can be derived from density. However, in this case, we observe the unnecessary weight generated near the green object surface (See Fig. 11(b) NeuS). It can be explained as a bias problem caused by inconsistency between the volume rendering weight and the SDF implicit surface. Inconsistency happened because there is a lack of constraint between the color field and

the geometry field (Chen et al., 2023). As a result, due to the bias problem, unnecessary points are sampled while thin structure (orange object) is ignored.

However, as shown in Fig. 11(b) $H_2O$-SDF w/o OGS, the introduction of the object surface field (OSF) shows its efficacy in alleviating the bias problem. The OSF learns a tight SDF surface and provides 3D supervision to the object surface and empty space, acting as a regularizer to geometry and volume rendering weight. Fig. 8 also demonstrates this effect, showing a reduction in unnecessary weight on the empty space.

We observe that the density-based sampling strategy still slightly over-emphasizes the already well-reconstructed room layout regions in the initial stage. To address this issue, we propose an OSF-guided sampling strategy to prioritize object surfaces exhibiting more intricate structures compared to room layouts. The probability of sampling is computed based on $w(\mathbf{x}) \cdot osf(\mathbf{x})$, where $\mathbf{x}$ is a spatial position. As illustrated in Fig. 11(b) $H_2O$-SDF, this strategy enhances the effectiveness of the OSF, inducing more densely sampled in regions where object surface probability and volume rendering weight are both high. Especially, it makes the learning process more focused on OSF and SDF collaboratively guiding each other, thereby improving the process of SDF capturing high-frequency detail.

**Refinement Loss Function Details** In this section, we provide more details of the refinement function loss. The point mask $\mathbb{1}_o(\mathbf{x}_j)$ in Eq. 5 is defined as:

$$\mathbb{1}_o(\mathbf{x}_j) = \begin{cases} 1, & osf(\mathbf{x}_j) \geq \theta \\ 0, & osf(\mathbf{x}_j) < \theta \end{cases} \tag{13}$$

where $\theta$ is a hyper-parameter to estimate whether input points belong to an object surface or not, and we set $\theta = 0.5$.

## A.3 ADDITIONAL EXPERIMENTS

**Curves of Standard Deviation** We additionally provide the standard deviation curve of the SDF during the learning process. For training the SDF network, NeuS introduces the probability density function to apply the volume rendering method and is defined as follows: $\phi_s(x) = se^{-sx}/(1 + e^{-sx})^2$. Here, $1/s$ represents the standard deviation of $\phi_s(x)$. During the network training converges, $1/s$ gradually decreases and tends towards zero, indicating a higher level of certainty and accuracy in the learned SDF representation. As shown in Fig. 12, our method exhibits near-zero standard deviation values compared to NeuS with normal and segmentation priors. This visualization result demonstrates that our method provides a robust geometry cue, enabling the effective learning of SDF.

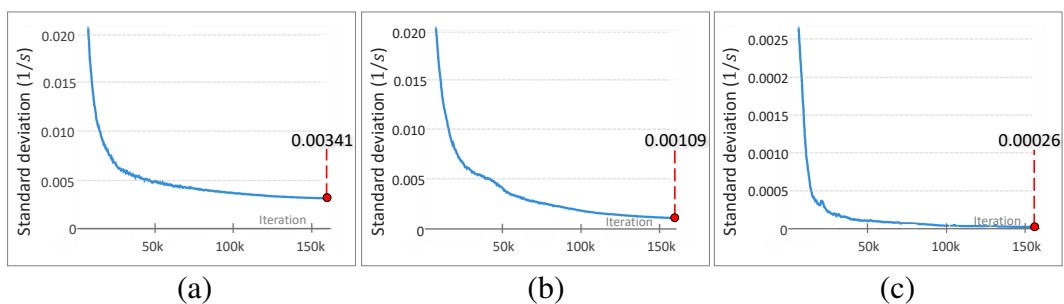

Figure 12: **Curve of Standard Deviation** Illustrations of (a) NeuS with normal priors, (b) NeuS with normal and segmentation priors, and (c) our method.

**Additional Analysis on the Object Surface Learning** To further analyze the effectiveness of our method, we provide additional experimental results that specifically focus on large object regions and layout regions pass through by empty space regions between multiple objects (See Fig. 13). As illustrated in Fig. 13(b), NeuS with a normal prior exhibits inconsistencies between the SDF implicit surface and the rendered surface. This inconsistency suggests the presence of a bias problem, highlighting the fact that simply incorporating geometry priors to guide the SDF representation does

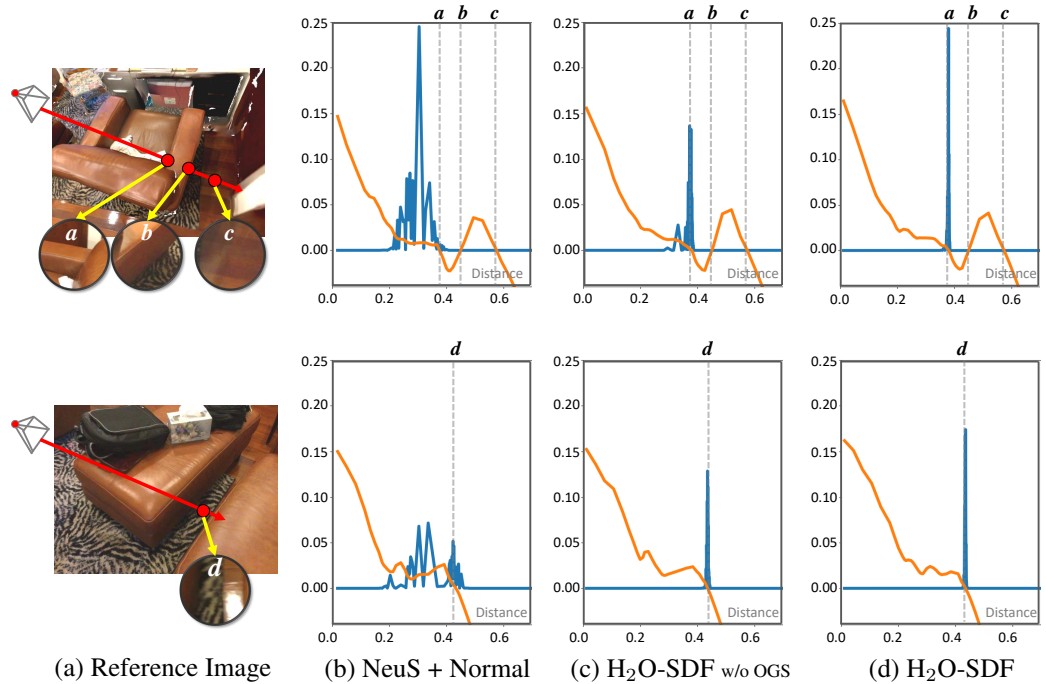

(a) Reference Image    (b) NeuS + Normal    (c) H$_2$O-SDF w/o OGS    (d) H$_2$O-SDF

Figure 13: **Additional Analysis on the Object Surface Learning** We followed the same visualization approach used in Fig. 8. In the case of the 1$^{st}$ row, the ray hits the surface of the sofa and the floor. In the case of the 2$^{nd}$ row, the ray passes through empty space between multiple objects while hitting the surface of the floors.

| HSL | | | OSL | | | | Accu.↓ | Comp.↓ | Prec.↑ | Recall↑ | F-score↑ |
|---|---|---|---|---|---|---|---|---|---|---|---|
| NeuS | Normal | Reweight | 2D | 3D | Ref | OGS | | | | | |
| ✓ | - | - | - | - | - | - | 0.132 | 0.114 | 0.376 | 0.334 | 0.353 |
| ✓ | ✓ | - | - | - | - | - | 0.040 | 0.042 | 0.794 | 0.742 | 0.766 |
| ✓ | ✓ | ✓ | - | - | - | - | 0.041 | 0.041 | 0.802 | 0.751 | 0.776 |
| ✓ | ✓ | ✓ | ✓ | - | - | - | 0.040 | 0.041 | 0.807 | 0.755 | 0.779 |
| ✓ | ✓ | ✓ | ✓ | ✓ | - | - | 0.040 | 0.041 | 0.809 | 0.760 | 0.784 |
| ✓ | ✓ | ✓ | ✓ | ✓ | ✓ | - | 0.039 | 0.041 | 0.817 | 0.762 | 0.787 |
| ✓ | ✓ | ✓ | ✓ | ✓ | ✓ | ✓ | 0.037 | 0.039 | 0.830 | 0.773 | 0.800 |

Table 5: Comprehensive ablation study of each component in H$_2$O-SDF.

not guarantee the accurate learning of the true SDF. In contrast, as illustrated in Fig. 13 (c) and (d), our method successfully builds the appropriate connection between the SDF implicit surface and the rendered surface. These results demonstrate the capability of object surface learning to accurately learn and represent the true SDF, surpassing the limitations of using geometry priors alone.

**Ablation Study** In this section, we report more detailed ablation study results for individual components in H$_2$O-SDF. The notations used in the table are as follows: (1) **NeuS** : Base model, (2) **Normal**: integration with normal prior, (3) **Reweight**: re-weighting scheme for Holistic Surface Learning defined in Sec. 3.1 (4) **2D**: 2D object surface loss $\mathcal{L}_{2d_{osf}}$, (5) **3D**: 3D object surface loss $\mathcal{L}_{3d_{osf}}$, (6) **Ref**: refinement loss $\mathcal{L}_{ref}$ and (7) **OGS**: Object Surface Field guided sampling strategy. Tab. 5 shows the progressive enhancement in performance with the integration of each component within the framework. This incremental improvement highlights the distinct effectiveness and contribution of each component to the overall model's efficacy.

**Normal Predictions** Fig. 14 and Tab. 6 present additional results of normal prediction, which compare the proposed method with other state-of-the-art neural implicit surface learning methods. As shown in Fig. 14, our method exhibits superior normal predictions with smoother and more

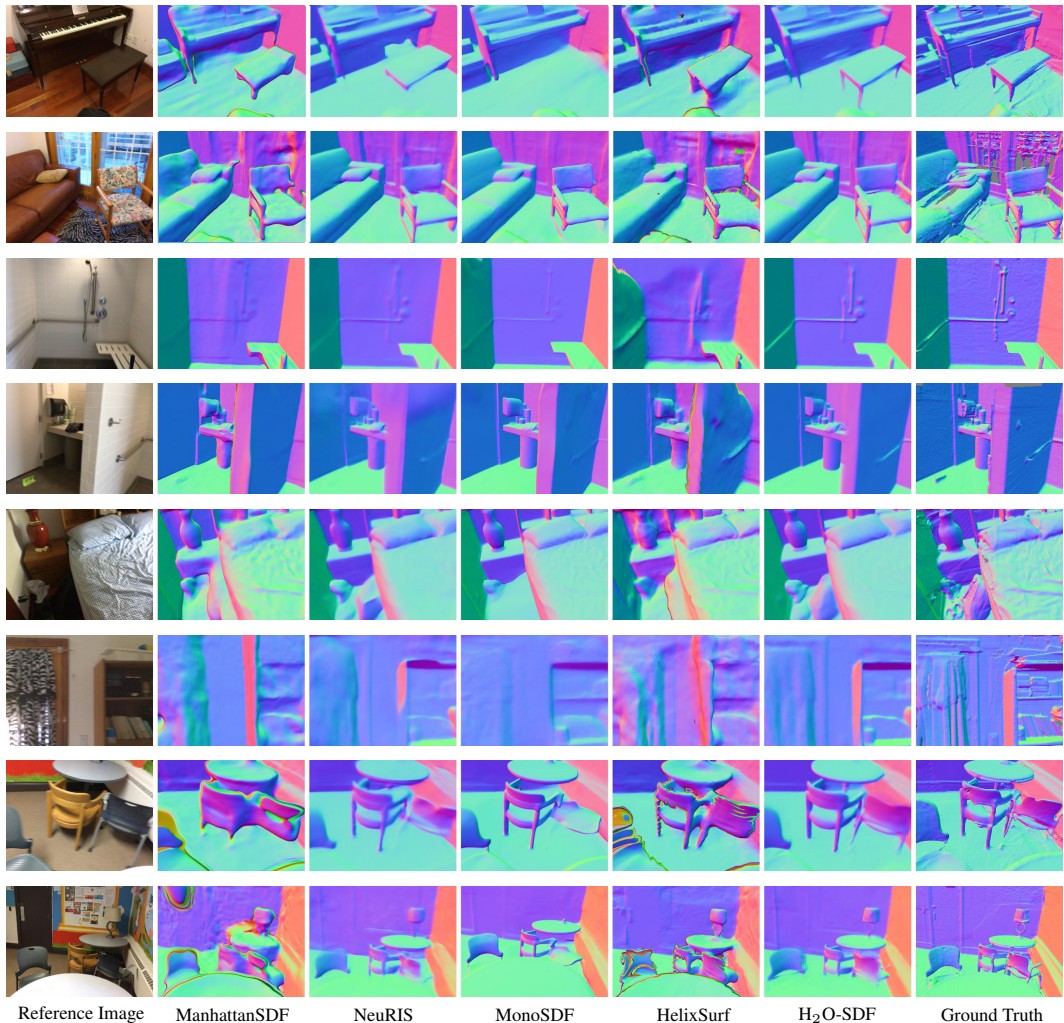

Figure 14: Additional Normal Prediction Results on ScanNet

| Methods | Mean↓ | Median↓ | RMSE↓ | 11.25 °↑ | 22.5 °↑ | 30 °↑ |
|---|---|---|---|---|---|---|
| ManhattanSDF | 19.14 | 10.82 | 29.43 | 52.86 | 74.4 | 80.96 |
| HelixSurf | 19.3 | 11.92 | 28.87 | 47.19 | 74.43 | 82.16 |
| NeuRIS | 15.21 | 8.77 | 23.19 | 61.46 | 79.76 | 85.42 |
| MonoSDF | 14.36 | 7.23 | 23.39 | 66.26 | 80.82 | 85.81 |
| **H$_2$O-SDF** | **13.48** | **6.96** | **21.73** | **66.94** | **81.82** | **86.90** |

Table 6: Quantitative Normal Comparisons on ScanNet

precise geometry. Moreover, Tab. 6 also demonstrates that our approach outperformed the existing methods in all normal metrics.

**3D Reconstructions** In this section, we provide additional qualitative and quantitative results of the selected 12 scenes on ScanNet. Results in Fig. 15 show that our proposed method achieves more smooth and fine-grained surface quality than other methods. According to the quantitative results (Tab. 7), our method outperforms the existing methods. However, for the *scene0603* and *scene0616*, our method performs slightly worse compared to NeuRIS (Wang et al., 2022a) and MonoSDF (Yu et al., 2022). This discrepancy can be attributed to the absence of ground truth meshes in certain

| Method | Scene0002_00 | | | | | Scene0012_00 | | | | |
|---|---|---|---|---|---|---|---|---|---|---|
| | Acc↓ | Comp↓ | Prec↑ | Recall↑ | F-score↑ | Acc↓ | Comp↓ | Prec↑ | Recall↑ | F-score↑ |
| ManhattanSDF | 0.053 | 0.070 | 0.665 | 0.561 | 0.608 | 0.044 | 0.048 | 0.736 | 0.707 | 0.722 |
| NeuRIS | 0.032 | 0.046 | 0.850 | 0.726 | 0.783 | 0.025 | 0.029 | 0.920 | 0.867 | 0.893 |
| MonoSDF | 0.061 | 0.065 | 0.692 | 0.606 | 0.646 | 0.04 | 0.048 | 0.710 | 0.638 | 0.672 |
| $H_2O$-SDF | 0.029 | 0.046 | 0.873 | 0.740 | 0.801 | 0.024 | 0.028 | 0.930 | 0.875 | 0.901 |

| Method | Scene0050_00 | | | | | Scene0084_00 | | | | |
|---|---|---|---|---|---|---|---|---|---|---|
| | Acc↓ | Comp↓ | Prec↑ | Recall↑ | F-score↑ | Acc↓ | Comp↓ | Prec↑ | Recall↑ | F-score↑ |
| ManhattanSDF | 0.058 | 0.059 | 0.707 | 0.642 | 0.673 | 0.055 | 0.053 | 0.639 | 0.621 | 0.630 |
| NeuRIS | 0.043 | 0.042 | 0.774 | 0.732 | 0.752 | 0.052 | 0.047 | 0.695 | 0.695 | 0.679 |
| MonoSDF | 0.041 | 0.053 | 0.717 | 0.625 | 0.668 | 0.038 | 0.030 | 0.856 | 0.882 | 0.870 |
| $H_2O$-SDF | 0.03 | 0.036 | 0.856 | 0.784 | 0.818 | 0.027 | 0.029 | 0.893 | 0.855 | 0.874 |

| Method | Scene0114_02 | | | | | Scene0279_00 | | | | |
|---|---|---|---|---|---|---|---|---|---|---|
| | Acc↓ | Comp↓ | Prec↑ | Recall↑ | F-score↑ | Acc↓ | Comp↓ | Prec↑ | Recall↑ | F-score↑ |
| ManhattanSDF | 0.153 | 0.083 | 0.513 | 0.503 | 0.508 | 0.105 | 0.068 | 0.577 | 0.588 | 0.583 |
| NeuRIS | 0.063 | 0.060 | 0.635 | 0.600 | 0.617 | 0.066 | 0.057 | 0.685 | 0.651 | 0.668 |
| MonoSDF | 0.202 | 0.091 | 0.469 | 0.470 | 0.469 | 0.086 | 0.059 | 0.642 | 0.636 | 0.639 |
| $H_2O$-SDF | 0.063 | 0.057 | 0.704 | 0.682 | 0.693 | 0.043 | 0.042 | 0.789 | 0.734 | 0.760 |

| Method | Scene0580_00 | | | | | Scene0603_00 | | | | |
|---|---|---|---|---|---|---|---|---|---|---|
| | Acc↓ | Comp↓ | Prec↑ | Recall↑ | F-score↑ | Acc↓ | Comp↓ | Prec↑ | Recall↑ | F-score↑ |
| ManhattanSDF | 0.104 | 0.062 | 0.616 | 0.650 | 0.632 | 0.143 | 0.066 | 0.583 | 0.620 | 0.601 |
| NeuRIS | 0.065 | 0.052 | 0.671 | 0.667 | 0.669 | 0.039 | 0.049 | 0.800 | 0.693 | 0.743 |
| MonoSDF | 0.0381 | 0.048 | 0.756 | 0.700 | 0.725 | 0.154 | 0.089 | 0.503 | 0.494 | 0.499 |
| $H_2O$-SDF | 0.031 | 0.034 | 0.863 | 0.827 | 0.845 | 0.043 | 0.049 | 0.784 | 0.689 | 0.734 |

| Method | Scene0616_00 | | | | | Scene0617_00 | | | | |
|---|---|---|---|---|---|---|---|---|---|---|
| | Acc↓ | Comp↓ | Prec↑ | Recall↑ | F-score↑ | Acc↓ | Comp↓ | Prec↑ | Recall↑ | F-score↑ |
| ManhattanSDF | 0.072 | 0.098 | 0.521 | 0.431 | 0.472 | 0.132 | 0.051 | 0.506 | 0.646 | 0.568 |
| NeuRIS | 0.048 | 0.057 | 0.712 | 0.614 | 0.659 | 0.067 | 0.057 | 0.731 | 0.695 | 0.713 |
| MonoSDF | 0.036 | 0.056 | 0.817 | 0.592 | 0.686 | 0.122 | 0.057 | 0.493 | 0.610 | 0.545 |
| $H_2O$-SDF | 0.040 | 0.051 | 0.730 | 0.615 | 0.667 | 0.049 | 0.047 | 0.771 | 0.726 | 0.748 |

| Method | Scene0625_00 | | | | | Scene0721_00 | | | | |
|---|---|---|---|---|---|---|---|---|---|---|
| | Acc↓ | Comp↓ | Prec↑ | Recall↑ | F-score↑ | Acc↓ | Comp↓ | Prec↑ | Recall↑ | F-score↑ |
| ManhattanSDF | 0.056 | 0.076 | 0659 | 0.613 | 0.635 | 0.088 | 0.055 | 0.675 | 0.640 | 0.657 |
| NeuRIS | 0.027 | 0.025 | 0.885 | 0.882 | 0.883 | 0.043 | 0.041 | 0.785 | 0.734 | 0.759 |
| MonoSDF | 0.031 | 0.036 | 0.814 | 0.785 | 0.799 | 0.046 | 0.047 | 0.723 | 0.665 | 0.693 |
| $H_2O$-SDF | 0.019 | 0.017 | 0.975 | 0.986 | 0.981 | 0.043 | 0.039 | 0.800 | 0.755 | 0.776 |

Table 7: **Quantitative Comparisons of Individual Scenes on ScanNet** For a fair comparison, we used the mesh provided officially by ScanNet as the ground truth. Therefore, the official results of each model may slightly differ from the results we measured.

| Time | ManhattanSDF | NeuRIS | MonoSDF | $H_2O$-SDF |
|---|---|---|---|---|
| Training (hours) | 5.2 | 4.5 | 21.5 | 4.5 |
| Inference (seconds) | 30 | 21 | 35 | 21 |

Table 8: Comparison of training and inference time

areas, as shown in Fig. 15 (see 5[th] and 6[th] row). Moreover, as demonstrated in the qualitative results Fig. 15, our method exhibits better reconstruction results for both scenes as well (see 5[th] and 6[th] row).

**Training and Inference Time** In Tab. 8, we present a comparison of training and inference times against baseline models. Our model demonstrates comparable speed to NeuRIS, outpacing the other models in terms of efficiency.

**Object Mesh Extraction** To demonstrate the potential applications of our trained object surface field (OSF), we extracted object mesh from our final reconstruction (See Fig. 16). To extract object mesh, we find the intersection of the signed distance function (SDF) and OSF where the SDF is near the zero-level set ($|d(\mathbf{x})| < \theta_d$) and OSF is above a certain threshold ($osf(\mathbf{x}) > \theta_{osf}$). Then, the intersection of the SDF remains and the rest of the SDF is filtered out. Then the mesh extraction from SDF is accomplished using the Marching Cubes algorithm (Lorensen & Cline, 1998). The results show the effectiveness and possibility of OSF in extracting object meshes.

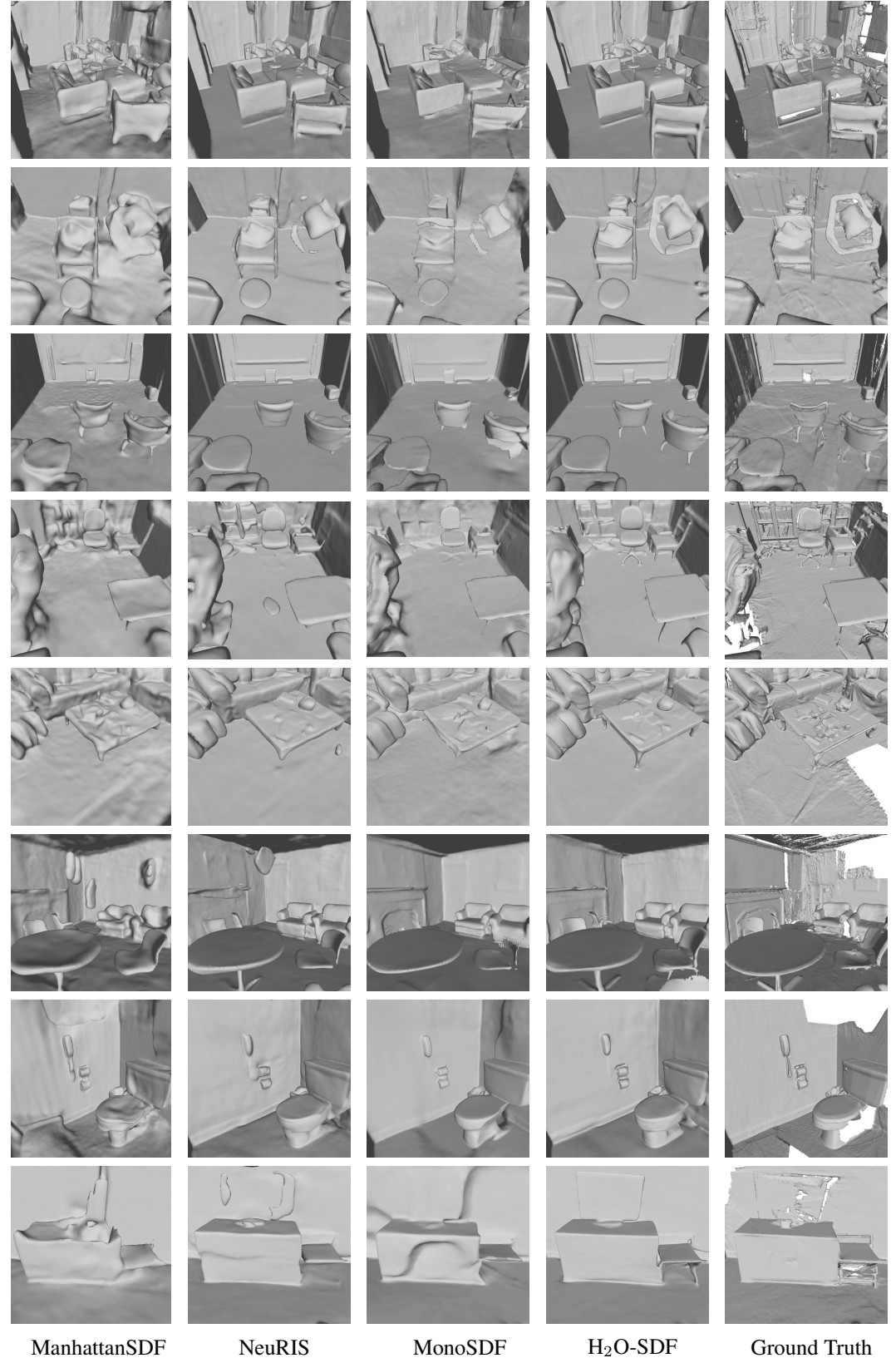

ManhattanSDF   NeuRIS   MonoSDF   $H_2O$-SDF   Ground Truth

Figure 15: Additional 3D Reconstruction Results on ScanNet

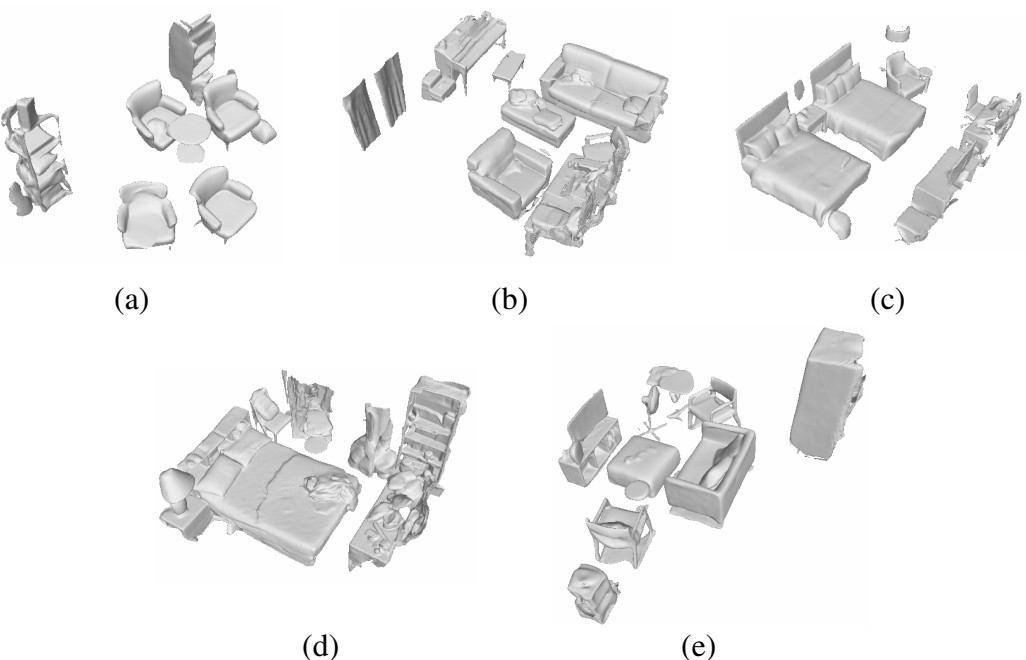

(a)          (b)          (c)

(d)          (e)

Figure 16: **Object Mesh Extraction** The results show the object meshes extracted from 5 ScanNet scenes.

