# OpenReview forum: "H2O-SDF: Two-phase Learning for 3D Indoor Reconstruction using Object Surface Fields"
_ICLR.cc/2024/Conference — ICLR 2024 spotlight_

### Official Review · Reviewer_ZNfu · 2023-10-31

**Soundness:** 3 good
**Presentation:** 2 fair
**Contribution:** 3 good
**Rating:** 6
**Confidence:** 4

**Summary:**

The paper proposes follow-up method to the SDF-based NeRF-like methods for indoor reconstruction, with a focus to improve geometry on objects in the second phase of optimization. The main novelty is the introduction of the auxiliary representation of Object Surface Field (OSF), which is activated on object surfaces. OSF can learned with 2D supervision of instance segmentation, as well as a loss in 3D which jointly constrain the SDF field and OSF field, bringing about zeros of SDF around object surfaces, leading to improved reconstruction on detailed high-frequency object parts. The method is evaluated against baseline SDF-based NeRF-like methods, on scenes including ScanNet and Replica.

**Strengths:**

[1] The proposal of using instance segmentation as additional input to the pipeline, as well as designing effective supervision signals with input segmentation.

Despite it is not new to use additional signals to the task (e.g. monocular normal and depth supervision for 2D-3D consistency, sparse points to supervise local SDF values, and using semantics and planar assumptions improve geometry of layouts), the paper is one of the first to demonstrate the usage of instance segmentation to improve fine geometry. More importantly, the paper does so in a non-trivial way, by introducing OSF to explicitly evolve object surfaces using a 2D loss between the input segmentation and OSF as well as a 3D loss between OSF and SDF.

[2] Illustration of the relationship of OSF and SDF (and the gradients), and the use of the OSF to drive SDF.

The paper provides informative illustration in Fig. 4 and related text on the relationship of OSF and SDF and how does the optimization of OSF loss drive SDF towards zero point around surfaces. The illustration using examples and 1D figures is clear and supports the motivation of the design of OSF.

[3] Extensive evaluation of the proposed method against baseline methods, and on more than one datasets.

**Weaknesses:**

[1] Clarification on OSF. Despite the good illustration of OSF as mentioned above, extra clarification is urgently needed to explain the motivation of the mathematical form of the 3D OSF loss (Equ. 2), and details in Fig. 4.

Specifically, despite Fig. 4 explains how the gradients of the 3 loss drives SDF to form a zero points around surfaces, and paper does not provide intuitive explanation on (a) why the various terms of the loss in Equ. 2 are designed as they are, (b) how \gamma controls the steepness of the function, how it matters and how $\gamma$ is picked (better with illustrations similar to Fig. 4). Additionally, it is not clear that, between Fig. 4 (a) and (c), why different d(x) lead to identical $\sigma_\gamma(x)$. Without clarifying the issues it is difficult to understand why OSF and the losses are designed the way they are, despite being proven effective.

[2] Demonstration of applying the proposed OSF and losses general SDF-like NeRF-based methods. The proposed OSF and losses should theoretically be applicable to all of the baselines methods as simple drop-in, but somehow the paper decides to compare against its own vanilla baseline. Is it possible to apply to other existing methods to better showcase the general nature of the proposed method, and how effective will it be?

[3] Limited scenes to evaluate. The main evaluation is done on ScanNet, with limited qualitative results on Replica. However ScanNet is known to have image quality issues. Why is the method not evaluated and compared on alternative datasets including Replica, Tanks & Temples, etc, as is done in other papers like MonoSDF? Without the additional evaluation, it is difficult to decide the generalization of the method across various indoor scenes.

[4] Additional comparison. One less important thing to add is, potential comparison with I^2-SDF. Despite I^2-SDF is based on additional geometry supervision signals, the goal is aligned with the proposed method, and it showcases similar improvement in fine detailed objects. It would be beneficial to add comparison to I^2-SDF to inspire the discussion on optimal strategy to improve reconstruction of high frequency signals in indoor geometry.

**Questions:**

Please see the Weakness section for questions to address.

---

> ### Author Response · Authors · 2023-11-20
> **Response to Reviewer ZNfu (Part 1/2)**
>
> We'd like to thank the reviewer for their time in making a detailed review with constructive feedback. We have numbered the weaknesses arisen as (Wx) and the questions arisen as (Qx) to ease the follow-up process where needed.
>
> (W1) *Clarification on OSF. Despite the good illustration of OSF as mentioned above, extra clarification is urgently needed to explain the motivation of the mathematical form of the 3D OSF loss (Equ. 2), and details in Fig. 4.
> Specifically, despite Fig. 4 explains how the gradients of the 3 loss drives SDF to form a zero points around surfaces, and paper does not provide intuitive explanation on (a) why the various terms of the loss in Equ. 2 are designed as they are, (b) how \gamma controls the steepness of the function, how it matters and how is picked (better with illustrations similar to Fig. 4). Additionally, it is not clear that, between Fig. 4 (a) and (c), why different d(x) lead to identical. Without clarifying the issues it is difficult to understand why OSF and the losses are designed the way they are, despite being proven effective.*
>
> (W_A1) Acknowledging the reviewer's feedback, we have prepared in the common response section a more detailed explanations of OSF and its corresponding loss functions.
>
> Addressing the reviewer's specific questions,
>
> (a) Equation 2 incorporates a loss term with dual components: one targeting the room layout and another for objects with intricate details. The room layout component tends to converge quickly due to a "smoothness bias". The object component, rich in high-frequency details, evolves differently. The design rationale behind Equation 2 is elaborated in the subsequent paragraphs following its introduction. The object region's loss term initially appears as depicted in Figure 4 (a). However, as training progresses, our Object Surface Field (OSF) evolves into the form shown in Figure 4 (e). Section 3.2, 'Mutual Induction of OSF and SDF', mathematically expounds this transition, focusing on the influence of the gradients of 3d_osf_loss on both OSF and the Signed Distance Function (SDF).
>
> (b) The 3d_osf_loss in our framework is engineered to modulate the OSF from 0 to 1, correlating with the SDF's transition from positive to negative values. The parameter gamma governs the rate at which OSF's value alters. Through experimental evaluations with varying gamma values, we observed that a lower gamma, indicating a gentler slope, enables the OSF to learn a more expansive region on the object. Conversely, a higher gamma, resulting in a steeper slope, restricts the OSF to a narrower region, potentially undermining its effectiveness in guiding the SDF. Based on these experiments, we have optimally selected a gamma value of 1.
>
> (c) Figure 4 (a) illustrates the scenario in which a well-trained SDF directs the OSF to accurately learn the surface field, particularly in general cases such as bulky furniture. On the other hand, Figure 4 (c) shows a situation where the OSF is effectively trained, possibly with additional inputs like an object mask, but the SDF has not yet accurately represented thinner structures. These examples are intended to demonstrate the reciprocal influence of OSF and SDF in varied training contexts.
>
> Acknowledging the reviewer's feedback, we will provide extra clarification to better explain the motivation of the mathematical form of the 3D OSF loss and other related concepts in the final paper submission.
>
> (W2)
> *Demonstration of applying the proposed OSF and losses general SDF-like NeRF-based methods. The proposed OSF and losses should theoretically be applicable to all of the baselines methods as simple drop-in, but somehow the paper decides to compare against its own vanilla baseline. Is it possible to apply to other existing methods to better showcase the general nature of the proposed method, and how effective will it be?*
>
> (W_A2) The reviewer's observation is valid as our Object Surface Field (OSF) can be seamless integrated with NeuS-based methods. However, we opted to incorporate our OSF into our enhanced baseline model, particularly after the initial stage of Holistic Surface Learning. This design choice is a strategic improvement over traditional baseline models like NeuRIS. While NeuRIS implements a normal prior and selectively utilizes normals based on their reliability, our Holistic Surface Learning goes a step further. It dynamically adjusts the utilization of both color and normal priors, contingent on the confidence level of the normal information. Consequently, we applied our Object Surface Learning to the refined Signed Distance Function (SDF) emerging from Holistic Surface Learning, rather than directly integrating it into the basic baseline models. This methodological choice was aimed at leveraging the advanced features of our enhanced model for more effective learning and integration of the OSF.

---

> > ### Author Response · Authors · 2023-11-20
> > **Response to Reviewer ZNfu (Part 2/2)**
> >
> > (W3) *Limited scenes to evaluate. The main evaluation is done on ScanNet, with limited qualitative results on Replica. However ScanNet is known to have image quality issues. Why is the method not evaluated and compared on alternative datasets including Replica, Tanks & Temples, etc, as is done in other papers like MonoSDF? Without the additional evaluation, it is difficult to decide the generalization of the method across various indoor scenes.*
> >
> > (W_A3) We agree with the reviewer's point. We couldn't not include our experimental results in the original submission due to page limit constraint but we now present additional experiments on the Replica dataset that include a comparision with MonoSDF as shown in the common commentions section. These experimental results indicate that our method is well generalizable across various indoor scenes.
> >
> > (W4) *Additional comparison. One less important thing to add is, potential comparison with I^2-SDF. Despite I^2-SDF is based on additional geometry supervision signals, the goal is aligned with the proposed method, and it showcases similar improvement in fine detailed objects. It would be beneficial to add comparison to I^2-SDF to inspire the discussion on optimal strategy to improve reconstruction of high frequency signals in indoor geometry.*
> >
> > (W_A4)
> > Our experimental findings address the reviewer's point well. The reviewer rightly points out that I^2-SDF aims to refine fine details in objects, albeit with the aid of additional supervision signals. I^2-SDF employs a "bubble loss", designed to minimize the Signed Distance Function (SDF) value to zero for 3D points back-projected from depth maps. This approach parallels the concept of View-aware SDF loss in Geo-NeuS, which similarly endeavors to drive the SDF value of points, derived from Multi-View Stereo (MVS), to zero.
> >
> > We explored the direct application of the View-aware SDF loss to the SDF derived from the MVS point cloud. However, our findings revealed that our method, which imparts indirect supervision to the SDF via the Object Surface Field (OSF), yielded superior results. This improvement can be attributed to the inherent noise often found in MVS point clouds, which may not align with the ground truth. Directly forcing the SDF values based on these noisy points can inadvertently degrade the quality of the final mesh. Thus, our method, which leverages the OSF as a mediating layer for SDF supervision, proves to be more effective in enhancing mesh quality, especially in scenarios where the input data may be imperfect.

---

> > > ### Author Response · Authors · 2023-11-23
> > > **A Gentle Reminder**
> > >
> > > Dear Reviewer ZNfu,
> > >
> > > Thank you for your time and efforts in reviewing our paper.
> > > We would like to gently remind you that the deadline for the discussion period is approaching.
> > > We sincerely hope that our responses and the results of our supporting experiments have clarified your concerns.
> > > We are looking forward to your feedback.
> > > Thank you so much!
> > >
> > > Authors

---

### Official Review · Reviewer_rsDk · 2023-11-01

**Soundness:** 3 good
**Presentation:** 2 fair
**Contribution:** 2 fair
**Rating:** 6
**Confidence:** 5

**Summary:**

The paper proposes a neural 3D indoor reconstruction framework to reconstruct 3D mesh of indoor scenes with a volume rendering framework. The key motivation of this paper is to decouple the learning of the layout and object with two stages. In the first stage, the layout of the scene is trained with an uncertainty-aware rendering loss function on both color and normal prediction. In the second stage, a new term named Object surface field (OSF) is introduced to measure the object occupancy of a 3D point, and authors demonstrate how SDF will facilitate SDF with the presented mutual induction. Extensive experiments on ScanNet have showcased the effectiveness of the proposed framework over different state-of-the-art (SOTA) methods.

**Strengths:**

(1) The motivation to decouple the learning of layout and object into two stage is straightforward and clear. The layout contains planar areas and the objects may have more high-frequency signals, thus may have different pace of convergence.

(2) The introduction of OSF is novel, and how the OSF can be transformed back to SDF and assist its representation is technically sound.

(3) Experiments on ScanNet have shown the advantages of proposed components of the method.

**Weaknesses:**

(1) The major concern for me is that of the technical impact of this work is limited by introducing a normal estimation network [1] which is also trained on ScanNet, to provide pseudo groundtruth normal and uncertainty during training. This cannot ensure fairness among baseline comparison and highly constraints the generalizability of the proposed method onto different benchmarks. A fair setting would be replace this network with another model or method which is pretrained on other datasets, or alternatively, test this method onto other indoor datasets such as 7-Scenes. This will significantly improve the fairness and technical impact of this work.

(2) In the supplementary material, authors present that they apply the OSF-based Filtering during reconstruction. I am curious about where does the major improvement of OSF comes from, either the proposed osf loss or the filtering. Authors are expected to conduct ablation study about this to make the contribution more convincing.

(3) Minor: The presentation can be further improved, and there exists noticeable typos in the submission such as Table 2.

**Questions:**

I appreciate the motivation of the design of this paper, however the use of a model seen on the same dataset limits the value of the proposed method. I would consider to improve my rating if my concerns listed in the weaknesses part can be well addressed.

---

> ### Author Response · Authors · 2023-11-19
> **Response to Reviewer rsDK**
>
> We'd like to thank the reviewer for their time in making a detailed review with constructive feedback. We have numbered the weaknesses arisen as (Wx) and the questions arisen as (Qx) to ease the follow-up process where needed.
>
> (W1) *The major concern for me is that of the technical impact of this work is limited by introducing a normal estimation network [1] which is also trained on ScanNet, to provide pseudo groundtruth normal and uncertainty during training. This cannot ensure fairness among baseline comparison and highly constraints the generalizability of the proposed method onto different benchmarks. A fair setting would be replace this network with another model or method which is pretrained on other datasets, or alternatively, test this method onto other indoor datasets such as 7-Scenes. This will significantly improve the fairness and technical impact of this work.*
>
> (W_A1) We value the reviewer's insightful feedback. The concern regarding the use of a normal estimation network trained on ScanNet and its implications for the fairness of balinese comparisons is indeed a valid point. We thoroughly considered this aspect prior to our submission but we still opted to proceed along the path of using ScanNet based pre-trained model, noting that numerous recent studies in indoor scene reconstruction using neural implicit representations have adopted similar approaches, leveraging various pretrained models as foundational models. For example, ManhattanSDF incorporates a segmentation model trained using ScanNet, and NeuRIS, akin to our approach, employs the same normal estimation model, also trained on ScanNet, but excludes test scenes from its dataset. Due to constraints in paper length, we were unable to include all experimental results at full extent that substituted our network with a different model or method trained on alternative datasets, or to extend testing of our method to other indoor datasets as suggested by the reviewer.
>
> **Table 1: Comparison results for Replica**
>
> | Model    | Accu.↓ | Comp.↓ | Prec.↑ | Recall↑ | F-score↑ |
> |----------|--------|--------|--------|---------|----------|
> | MonoSDF  | 0.020  | 0.017  | 0.951  | 0.923   | 0.934    |
> | H2O-SDF  | 0.016  | 0.025  | 0.983  | 0.935   | 0.957    |
>
>
> **Table 2: Comparison results for 7-Scenes**
>
> | Model         | Accu.↓ | Comp.↓ | Prec.↑ | Recall↑ | F-score↑ |
> |---------------|--------|--------|--------|---------|----------|
> | ManhattanSDF  | 0.112  | 0.132  | 0.351  | 0.326   | 0.336    |
> | NeuRIS        | 0.133  | 0.132  | 0.405  | 0.424   | 0.410    |
> | H2O-SDF       | 0.128  | 0.129  | 0.416  | 0.469   | 0.447    |
>
> Acknowledging the reviewer’s valid concern, we are now presenting additional experiments using the Replica and 7-scenes datasets. In our first experiment, detailed in Table 1, we utilized an Omnidata-based surface normal estimation to test our model with a surface normal estimation model pretrained on a different dataset. This directly responds to the reviewer's suggestion of replacing our ScanNet-based network. In our second experiment, shown in Table 2, we applied our model to the 7-scenes dataset using ScanNet-based surface normal estimation, responding to the suggestion by the reviewer to test our method on other indoor datasets.
>
> The results, as illustrated in Tables 1 and 2,
> highlight our method's robust performance and its ability to generalize effectively, irrespective of the pre-trained model used for normal estimation or the specific domain of application. In the final version of our paper, we intend to include these additional experimental results either in the main body or in an appendix.
>
> (W2) *In the supplementary material, authors present that they .... more convincing.*
>
> (W_A2) In the supplementary material, the Object Mesh Extraction via OSF-based Filtering serves merely as an example to demonstrate the potential applications of our trained OSF. Note that this filtering process is not a core component of our proposed method. Hence, we do not anticipate a notable enhancement in performance during reconstruction through OSF-based filtering. To avoid any confusion for the reader, we will explicitly clarify this distinction in the final version of the paper.
>
> (W3) *Minor: .... Table 2.*
>
> (W_A3)
> We will enhance the readability of our paper by correcting all typos, including those in Table 2.* and elsewhere in the submission.
>
> (Q1) *I appreciate the motivation of the design of this paper, however the use of a model seen on the same dataset limits the value of the proposed method. I would consider to improve my rating if my concerns listed in the weaknesses part can be well addressed.*
>
> (Q_A1) We hope that our response to Weakness 1 (W_A1) adequately addresses the concern raised by the reviewer. We would be grateful if the reviewer could reconsider and, if deemed appropriate, improve the rating based on these amendments.

---

> > ### Author Response · Authors · 2023-11-23
> > **A Gentle Reminder**
> >
> > Dear Reviewer rsDK,
> >
> > Thank you for your time and efforts in reviewing our paper.
> > We would like to gently remind you that the deadline for the discussion period is approaching.
> > We sincerely hope that our responses and the results of our supporting experiments have clarified your concerns.
> > We are looking forward to your feedback.
> > Thank you so much!
> >
> > Authors

---

### Official Review · Reviewer_ypMg · 2023-11-01

**Soundness:** 3 good
**Presentation:** 3 good
**Contribution:** 3 good
**Rating:** 8
**Confidence:** 4

**Summary:**

This paper proposes a two-phase learning approach named H2O-SDF that combines both holistic surface learning and object surface learning, for 3D reconstruction in indoor environments.

The main contributions are: 1) a two-phase learning framework that balances between the reconstruction of global room geometry and local object details. 2) Introduction of Object Surface Field (OSF), a new concept designed to address the vanishing gradient problem suffered by SDF, which hinders the reconstruction of high-frequency details. The authors also introduce an OSF-guided sampling strategy to prioritize object surfaces in the sampling process.

**Strengths:**

1. This paper tackle an important issue in the field of 3D indoor scene reconstruction — the difficulty of preserving the overall geometry while capturing intricate object details. It introduces a two-phase learning approach, which has not been explored before.
2. The OSF concept is new and shows promising results in handling the inherent vanishing gradient issue in the learning process.
3. It is an interesting idea to use normal uncertainty as a guidance to re-weight normal and color loss, to adaptively moderate normal and color losses in both low-texture and texture-rich regions.
4. The submission appears to be well-organized with its ideas clearly articulated.
5. Experimental evaluations, together with ablation studies, confirm the effectiveness of H2O-SDF. The results show that the proposed solution outperforms existing state-of-the-art methods.

**Weaknesses:**

1. The explanation and exposition of some key, novel concepts, such as OSF, L2D_OSF, L3D_OSF, could be more thorough. There is insufficient mathematical detail on the OSF guided sampling strategy (although there is graphical illustration in the appendix A2, the explanation seems to be mostly a repetition of the main body). Strengths of the proposed formulation could be better appreciated by providing more detailed explanations and mathematical insights.

2. Running time: The paper does not provide specific details about the computational complexity or running time of the approach, for both training and inference.  It only states that all experiments were conducted on a single NVIDIA RTX 3090Ti GPU.

3. Comparison with more diverse data (this is more of a suggestion): While the paper compares favorably to state-of-the-art methods on the ScanNet dataset, it would strengthen the paper to include a broader range of data under different conditions, such as different indoor layout complexities, object variations etc.

**Questions:**

1. It would be interesting to find out to what extent this method relies on pre-trained models and priors, which might limit its application in environments where such models are not easily available.

2. Running time/computational complexity of the proposed method. Please refer to point 2 under Weaknesses.

---

> ### Author Response · Authors · 2023-11-20
> **Response to Reviewer ypMg**
>
> We'd like to thank the reviewer for their time in making a detailed review with constructive feedback. We have numbered the questions arisen as (Qx) to ease the follow-up process where needed.
>
> (Q1) *It would be interesting to find out to what extent this method relies on pre-trained models and priors, which might limit its application in environments where such models are not easily available.*
>
> (Q\_A1)
> We agree with the reviewer's insight that delving into the extent to which our method relies on pre-trained models and prior knowledge is indeed important. As our model draws cues for indoor scenes from readily available pre-trained models, we have confidence in its potential for broad generalizability across a range of generic and diverse indoor scenes. This assertion is further supported by our empirical findings, which include experiments conducted on additional datasets such as 7-scenes and Replica. We have elaborated on these results in the common comments section, and we cordially invite the reviewer to refer this section for a more comprehensive understanding. Furthermore, we intend to incorporate these findings into the final version of our paper for submission.
>
> (Q2) *Running time/computational complexity of the proposed method. Please refer to point 2 under Weaknesses.*
>
> (Q\_A2)
>
> Below, we present the average training and inference times obtained from our experiments on the 12 ScanNet scenes:
>
> |  Time                  | ManhattanSDF | NeuRIS | MonoSDF | H2O-SDF |
> |--------------------|--------------|--------|---------|---------|
> | Training (hours)    | 5.2     | 4.5     | 21.5    | 4.5     |
> | Inference (seconds) | 30      | 21      | 35        | 21       |
>
> We plan to include these results in the Appendix of our final paper submission.

---

> > ### Author Response · Authors · 2023-11-23
> > **A Gentle Reminder**
> >
> > Dear Reviewer ypMg,
> >
> > Thank you for your time and efforts in reviewing our paper.
> > We would like to gently remind you that the deadline for the discussion period is approaching.
> > We sincerely hope that our responses and the results of our supporting experiments have clarified your concerns.
> > We are looking forward to your feedback.
> > Thank you so much!
> >
> > Authors

---

### Official Review · Reviewer_93pm · 2023-11-10

**Soundness:** 3 good
**Presentation:** 2 fair
**Contribution:** 3 good
**Rating:** 6
**Confidence:** 3

**Summary:**

This paper proposed a two-phase framework (H2O-SDF) for 3D indoor scene reconstruction. In particular, the proposed method adopts a two-stage method, which consists of one-stage reconstruction for the scene layout followed by a second-stage reconstruction of the objects using NERF. The key contribution is to introduce the concept of the object surface field. The 2D and 3D object surface losses are introduced to estimate the SDF for fine object surface details. The experiments are conducted on ScanNet and show superior results compared with existing methods.

**Strengths:**

+ The method reconstruct the layout and the object separately and achieves very good reconstruction on the details of the objects.
+ The introduced OSF captures the occupancy of the surface of the 3D object.
+ The introduced two losses let the SDF captured more surface details.

**Weaknesses:**

-	2D object surface loss. Could it be explained as the loss between the rendered object masks and the ground truth masks? It would be great to make it clear that the proposed method actually requires object annotations.
-	It would be great to explain OSF with more details. Based on the description in the paper, it is quite similar to the absolute gradient field of the occupancy values. In particular, Eq. 3 actually enforces the OSF to have large values on the object defined by the 3D points.  In addition, 3D points provide strong prior on the details of the shapes. It would be great to provide the ablations study of using the point cloud with MVS images or not.
-	Experiments on ablations studies. It is not clear to the reviewer what model A, B, C are. It would be great to provide detailed explanations about those models.
-	For the second stage, it would be great to ablate whether all the losses have contributed to the final results. The proposed method adopts more accurate point cloud obtained from MVS images compared with monocular depth estimated from a single image. Those factors should be ablated to demonstrate the performance benefits from OSF and the sampling strategy not from the prior data

**Questions:**

- It would be great to elaborate more on the insight of OSF and also compared with existing density functions parameterised with the SDF values.
- The ablations studies are missing. Please demonstrate all the losses introduced in stage two all contributes to the improvement of the reconstruction. In addition the proposed method leverages the point cloud obtained from MVS. It would be great to show how these priors can influence the final performance.

---

> ### Author Response · Authors · 2023-11-20
> **Response to Reviewer 93pm (Part 1/3)**
>
> We'd like to thank the reviewer for their time in making a detailed review with constructive feedback. We have numbered the weaknesses arisen as (Wx) and the questions arisen as (Qx) to ease the follow-up process where needed.
>
> (W1) *2D object surface loss. Could it be explained as the loss between the rendered object masks and the ground truth masks? It would be great to make it clear that the proposed method actually requires object annotations.*
>
> (W_A1)
> Thank you for highlighting the lack of clarity in our explanation regarding the non-requirement of object annotations for the 2D object mask. In Section 3.2, Line 6 of "2D Object Surface Loss", we mentioned, "The 2D object mask is determined by a process which can be easily executed using a pre-trained model (Lambert et al., 2020)". However, we realize this could lead to confusion about whether annotations are necessary during the training of the segmentation model. In our final paper submission, we will explicitly state in Section 4.1, "Implementation Details", that "The 2D object mask is obtained using the MSeg segmentation method, employing its officially provided pre-trained model".
>
> The concept of 2D object surface loss is essentially the disparity between the volume-rendered result of the object surface field and the 2D object label (pseudo-ground truth), which is derived from a pre-trained segmentation model, MSeg. This approach enables us to acquire pseudo-ground truth labels without using object annotations during the training process that relies on this loss.
>
> (W2) *It would be great to explain OSF with more details. Based on the description in the paper, it is quite similar to the absolute gradient field of the occupancy values. In particular, Eq. 3 actually enforces the OSF to have large values on the object defined by the 3D points. In addition, 3D points provide strong prior on the details of the shapes. It would be great to provide the ablations study of using the point cloud with MVS images or not.*
>
> (W_A2)
> Please refer to the common response section a full detailed explanations of OSF and its corresponding loss functions.
>
> Quoting what is already mentioned in the common response section:
>
> ***"OSF essentially quantifies the likelihood (ranging from 0 to 1) of the object surface OSF(x) at each spatial point, enhancing the SDF’s ability to better capture fine geometric details and high-frequency surfaces, while ensuring a smooth room layout"***
>
> It implies that there are clear differences between OSF and absolute gradient field. That is, the Object Surface Field is bounded between 0 and 1, indicating the probability of an object's surface presence, whereas the absolute gradient field's values can range from 0 to infinity. Furthermore, our Object Surface Field is not merely the gradient field of an imperfectly trained Signed Distance Function (SDF) using color and normal information. Instead, it serves as an independent representation, enhancing the SDF by indicating regions it has yet to encapsulate, thereby acting as a crucial 3D geometric cue.
>
> Moreover, incorporating refine loss (ref_loss), which leverages the point cloud obtained from Multi-View Stereo (MVS), improves the model's capability in capturing intricate shapes. However, it is essential to emphasize that ref_loss plays a complementary role. Our model demonstrates state-of-the-art performance independently of the ref_loss component. The below experimental results support this claim.
>
> | Methods                   | Acc↓   | Comp↓  | Prec↑  | Recall↑ | F-score↑ |
> |---------------------------|--------|--------|--------|---------|----------|
> | NeuS                      | 0.179  | 0.208  | 0.313  | 0.275   | 0.291    |
> | ManhattanSDF              | 0.053  | 0.056  | 0.715  | 0.664   | 0.688    |
> | NeuRIS                    | 0.052  | 0.050  | 0.713  | 0.677   | 0.690    |
> | MonoSDF                   | 0.035  | 0.048  | 0.799  | 0.681   | 0.733    |
> | HelixSurf                 | 0.038  | 0.044  | 0.786  | 0.727   | 0.755    |
> |---------------------------|--------|--------|--------|---------|----------|
> | H2O-SDF w/o ref_loss   | 0.034  | 0.038  | 0.824  | 0.757   | 0.789    |
> | H2O-SDF                   | 0.032  | 0.0373 | 0.834  | 0.769   | 0.799    |

---

> ### Author Response · Authors · 2023-11-20
> **Response to Reviewer 93pm (Part 2/3)**
>
> (W3) *Experiments on ablations studies. It is not clear to the reviewer what model A, B, C are. It would be great to provide detailed explanations about those models.*
>
> (W_A3) We admit that we employed some unclear notations. In the final paper submission, we plan to employ more enhanced notations in the ablation study part. That is, here is a revised summary:
>
> In our ablation study, we compare various model configurations: Model A, Model B, Model C, and H2O-SDF. Model A, our baseline, is an adaptation of NeuS with the integration of normal loss. Model B represents a scenario where the model undergoes training solely through our first stage, Holistic Surface Learning (NeuS + Normal + re-weighting scheme), right up to the final epoch, but does not include the Object Surface Learning phase. Model C and H2O-SDF, on the other hand, adopt a two-phase learning process. In this approach, Object Surface Learning (OSL) is initiated after the completion of Holistic Surface Learning (HSL). Specifically, Model C explores the impact of conducting OSL without employing the OSF-guided sampling strategy (OGS).
>
> (W4) *For the second stage, it would be great to ablate whether all the losses have contributed to the final results. The proposed method adopts more accurate point cloud obtained from MVS images compared with monocular depth estimated from a single image. Those factors should be ablated to demonstrate the performance benefits from OSF and the sampling strategy not from the prior data*
>
> (W_A4) We report our ablation study results for each loss used during Object Surface Learning (i.e. 2nd stage of our method). The notations used in the table are as follows.
>
> - HSL: Holistic Surface Learning, i.e., color and normal reweighting scheme.
> - 2D: 2D object surface loss.
> - 3D: 3D object surface loss.
> - ref: refinement loss.
> - OGS: Object Surface Field guided sampling strategy.
>
> | Model                                 | Acc↓  | Comp↓ | Prec↑ | Recall↑ | F-score↑ |
> |---------------------------------------|-------|-------|-------|---------|----------|
> | Neus + HSF                            | 0.041 | 0.041 | 0.802 | 0.751   | 0.776    |
> | Neus + HSF + 2D                       | 0.040 | 0.041 | 0.807 | 0.755   | 0.779    |
> | Neus + HSF + 2D + 3D                  | 0.040 | 0.041 | 0.809 | 0.760   | 0.784    |
> | Neus + HSF + 2D + 3D + ref            | 0.039 | 0.041 | 0.817 | 0.762   | 0.787    |
> | Neus + HSF + 2D + 3D + ref + OGS      | 0.037 | 0.039 | 0.830 | 0.773   | 0.800    |
>
> Considering the page limit constraint, we will include this table in the Appendix of our final paper submission to clearly demonstrate the effectiveness of each loss.

---

> > ### Author Response · Authors · 2023-11-20
> > **Response to Reviewer 93pm (Part 3/3)**
> >
> > (Q1) *It would be great to elaborate more on the insight of OSF and also compared with existing density functions parameterised with the SDF values.*
> >
> > (Q_A1)
> > We thank the reviewer for this suggestion. In the final paper submission, to clearly show the insight of OSF, we will include the following explanation:
> > "It is revealed through several works that models tend to focus on low frequency regions during training, known as "smoothness bias." Therefore, we introduce the Object Surface Field (OSF), a novel representation that delivers three-dimensional spatial information to the model. This concept is interesting as it extends beyond existing 2D priors, functioning as a 3D geometry cue that delivers 3D spatial information to the SDF."
> >
> > Additionally, we hope that the answer (W_A2) to Weakness 2, which provided us valuable insights, will address the relation between SDF and OSF.
> >
> > (Q2) *The ablations studies are missing. Please demonstrate all the losses introduced in stage two all contributes to the improvement of the reconstruction. In addition the proposed method leverages the point cloud obtained from MVS. It would be great to show how these priors can influence the final performance.*
> >
> > (Q\_A2)
> > We report the ablation study results for each loss used in stage two, Object Surface Learning, in our answer to Weakness 4 (W\_A4). In addition, in addressing Weakness2 (W\_A2), we include the ablation study results for ref_loss, which utilizes the MVS point cloud.

---

> > > ### Author Response · Authors · 2023-11-23
> > > **A Gentle Reminder**
> > >
> > > Dear Reviewer 93pm,
> > >
> > > Thank you for your time and efforts in reviewing our paper.
> > > We would like to gently remind you that the deadline for the discussion period is approaching.
> > > We sincerely hope that our responses and the results of our supporting experiments have clarified your concerns.
> > > We are looking forward to your feedback.
> > > Thank you so much!
> > >
> > > Authors

---

### Author Response · Authors · 2023-11-20
**Common Response (Part 1/2)**

To address the common concern raised by some reviewers on the question of in what extent our method relies on pre-trained models and priors, and how generalizable our method is for other indoor datasets. We present additional experimental results using the Replica and 7-scenes datasets. In our first experiment, detailed in Table 1, we utilized an Omnidata-based surface normal estimation, testing our model based a surface normal estimation model pre-trained on a different dataset. In our second experiment, shown in Table 2, we applied our model to the 7-scenes dataset using ScanNet-based surface normal estimation, testing our method on other indoor datasets. All settings used in these experiment are kept consistent with those presented in our original submission.

**Table 1: Comparison results for Replica**

| Model    | Accu.↓ | Comp.↓ | Prec.↑ | Recall↑ | F-score↑ |
|----------|--------|--------|--------|---------|----------|
| MonoSDF  | 0.020  | 0.017  | 0.951  | 0.923   | 0.934    |
| H2O-SDF  | 0.016  | 0.025  | 0.983  | 0.935   | 0.957    |

**Table 2: Comparison results for 7-Scenes**

| Model         | Accu.↓ | Comp.↓ | Prec.↑ | Recall↑ | F-score↑ |
|---------------|--------|--------|--------|---------|----------|
| ManhattanSDF  | 0.112  | 0.132  | 0.351  | 0.326   | 0.336    |
| NeuRIS        | 0.133  | 0.132  | 0.405  | 0.424   | 0.410    |
| H2O-SDF       | 0.128  | 0.129  | 0.416  | 0.469   | 0.447    |


The results, as illustrated in Tables 1 and 2, clearly show our method's robust performance and its ability to generalize effectively, irrespective of the pre-trained model used for normal estimation or any specific domain of application. In the final paper submission, we intend to include these additional experimental results either in the main body or in an appendix to strengthen the quality of our final submission.

---

> ### Author Response · Authors · 2023-11-22
> **Common Response (Part 2/2)**
>
> To address the common request for a detailed explanation of Object Surface Field (OSF), we provide insights of OSF and a detailed explanation of each loss term.
>
> **a) Insight of OSF:**
>
> As demonstrated in Appendix A.2, the NeuS-based reconstruction method using only simple rendering losses suffers from ***the vanishing gradient problem*** in the high-frequency region. To address this issue, directly guiding the SDF with incomplete geometric prior knowledge can negatively impact the quality of reconstruction.
>
> Therefore, we introduce a new representation called Object Surface Field (OSF), which is a more interesting concept than the existing 2D priors as it serves as a 3D geometric clue conveying spatial information to the model. OSF essentially quantifies the likelihood (ranging from 0 to 1) of the object surface OSF(x) at each spatial point, enhancing the SDF’s ability to better capture fine geometric details and high-frequency surfaces, while ensuring a smooth room layout.
>
>
> **b) The detailed explanation of the representative loss terms of OSF:**
>
> To train the OSF, two essential loss terms are utilized: $\mathcal{L}\_{2d\\_osf}$ and $\mathcal{L}\_{3d\\_osf}$.
>
> **b-1) Detailed explanation of $\mathcal{L}_{2d\\_osf}$ :**
>
>  The 2D object surface loss is a preliminary step in obtaining the initial value of OSF. In more detail, the concept of the 2D object surface loss is essentially the disparity between the volume-rendered result of the object surface field $osf(\mathbf{r})$ along a ray $\mathbf{r}$ and the 2D object label (pseudo-ground truth) $\mathbb{1}_o(\mathbf{r})$, which is derived from a pre-trained segmentation model, MSeg.
>
> By minimizing the discrepancy between these two representations with multi-view consensus, $\mathcal{L}_{2d\\_osf}$ established the initial value of OSF.
>
> **b-2) Detailed explanation of $\mathcal{L}_{3d\\_osf}$ :**
>
>  The 3D object surface loss enables OSF to carefully learn the object surface from SDF, letting SDF capture high-frequency details under the guidance of OSF. Unlike $\mathcal{L}_{2d\\_osf}$, which uses the volume-rendered result, $\mathcal{L}\_{3d\\_osf}$ utilizes the probability of the object surface for each spatial point $\mathbf{x}\_{i}$ and the corresponding transformed SDF values by scaled sigmoid function $\sigma\_{\gamma}(\mathbf{x}\_{i})$.
>
> The $\mathcal{L}\_{3d\\_osf}$ comprises two parts, the object term and the room layout term. The room layout term $((1-\mathbb{1}_o(\mathbf{r})) \cdot osf(\mathbf{x}\_{i}))$ dictates that points along a ray, identified as the room layout by the 2D object mask, should have low object surface probabilities. Conversely, the object term $(\mathbb{1}_o(\mathbf{r}) \cdot osf(\mathbf{x}\_{i}) \cdot |osf(\mathbf{x}\_{i}) - \sigma\_{\gamma}(\mathbf{x}\_{i})|)$ addresses rays intersecting with object surfaces. In indoor scenes, such rays inevitably also pass through room layout area after intersecting an object. To distinguish between 3D points related to objects and room layout, we multiply $osf(\mathbf{x}\_{i})$ in the formula. Due to the room layout term and the context of multi-view settings, points corresponding to the room layout exhibit low osf values and are thus unaffected by the term $|osf(\mathbf{x}\_{i}) - \sigma\_{\gamma}(\mathbf{x}\_{i})|$.
>
> The expression $|osf(\mathbf{x}\_{i}) - \sigma\_{\gamma}(\mathbf{x}\_{i})|$ indicates that the object surface probability should transition from 0 to 1 as SDF values move from positive to negative, or from outside to inside the object. The scaled sigmoid function $\sigma_{\gamma}(\mathbf{x}\_{i})$ is defined as $\frac{1}{1 + \exp(\gamma \cdot d(\mathbf{x}\_{i}))}$, where $\gamma$ controls the steepness of this transition. Our experiments with different $\gamma$ values revealed that a smaller gamma, which corresponds to a less steep transition, allows the OSF to cover a broader area of the object. A larger $\gamma$, indicating a steeper transition, confines the OSF to a narrower region, which may reduce its ability to provide effective guidance to the SDF. This loss function allows us to convert 2D object mask data into a 3D geometric cue, which is the essence of our novel concept, the object surface field.

---

### Meta-Review · Area_Chair_sEa2 · 2023-12-06

**Metareview:**

This paper presents H2O-SDF for 3D indoor scene reconstruction. It uses a two-stage method, which consists of the reconstruction of the global scene surface (first stage) and the reconstruction of objects using NeRF (second stage). To this end, it introduces object surface fields (OSFs) to alleviate the vanishing gradient problem suffered by SDFs. The result clearly demonstrates the effectiveness of the proposed method.

The major strengths of the paper are:
(1) The proposed method overcomes an important issue of jointly preserving the overall geometry and capturing intricate object details.
(2) The introduction of OSF is novel, and its use with SDF is technically sound.
(3) Evaluation is comprehensive, and the experiment clearly showcases the strength of the proposed method.

On the other hand, the weaknesses are:
(1) Limited explanation about OSF. It would have been nicer with more mathematical details to explain its insights.
(2) The data used for the experiment is rather limited.

Overall, reviewers and AC are positive about the paper. While the weaknesses above were pointed out in the review and they were not fully addressed, the merit of the paper surpasses the negatives. The reviewers and AC read the rebuttal and took it into consideration to reach the final recommendation.

**Justification For Why Not Higher Score:**

There were remaining concerns about the limited explanation of OSF and the limited diversity of the data that were used for the experiment. The rebuttal addressed these issues to some extent; however, it did not fully address them.

**Justification For Why Not Lower Score:**

Although there were some negative concerns described above, the reviewers and AC clearly see the merit of the paper. In particular, the introduction of OSF and the two-stage approach to indoor scene reconstruction is interesting.

---

### Decision · Program_Chairs · 2024-01-16

Accept (spotlight)